# *CASCO-Agent*: COST-AWARE SIMULATION CONFIGURATION VIA SURROGATE-GUIDED AGENTS

## ABSTRACT

Configuring physics-based simulations requires balancing granularity against computational budget, a dilemma we term **C**ost-**A**ware **S**imulation-Based **C**onfiguration **O**ptimization (*CASCO*). Traditional methods, such as Bayesian optimization or manual expert design, often struggle with the curse of high dimensionality or fail to generalize. Large Language Models (LLMs) offer promise for automating such workflows but, as we show experimentally, lack inherent cost awareness and frequently propose inefficient configurations. While inference-time scaling can improve the exploration width to find cost-efficient configurations, it demands prohibitively many simulator queries. We propose **C**ost-**A**ware **S**imulation **C**onfiguration **O**ptimization Agent(*CASCO-Agent*), an agentic framework guiding inference-time scaling via lightweight surrogates. The surrogate here only predicts low-dimensional metrics (accuracy, cost) rather than complete physics fields. This enables easier training and flexible adaptation to data availability, e.g., using Gaussian Processes in data-scarce regimes or Neural Networks when data is abundant. In experiments across 3 typical PDE solvers (elliptic, parabolic, and hyperbolic), *CASCO-Agent* consistently outperforms Bayesian optimization and LLM-based baselines, achieving success rates comparable to inference-time scaling with a ground truth simulator without incurring expensive simulation overhead.

## 1 INTRODUCTION

Physics-based simulations are the backbone of modern engineering, playing critical roles in inverse design (Loonen et al., 2022; Jabbar et al., 2022) and control (Lawrence et al., 2024). In these pipelines, an outer optimization loop iteratively queries an inner simulator to conduct forward simulation, evaluate design objectives, and adjust control variables (Vlastelica et al., 2023; Molesky et al., 2018). This process creates a fundamental dilemma regarding the computational budget: overly coarse simulator configurations may yield misleading feedback that derails downstream tasks, while finer configurations offer precision but rapidly exhaust resources. Achieving the balance between sufficient precision and economical cost is an open challenge we term **C**ost-**A**ware **S**imulation-Based **C**onfiguration **O**ptimization (*CASCO*).

Traditional approaches to *CASCO* fall short of scalability. 1) **Brute force search** is computationally intractable for scanning high-dimensional parameter spaces. 2) **Bayesian Optimization (BO)** (Snoek et al., 2012; Yao et al., 2024) and evolutionary methods (Perera et al., 2023) offer potentially higher efficiency but struggle to generalize across varying environments or incorporate domain knowledge expressed in natural language. 3) **Expert manual design**, when available, is effective but labor-intensive, creating a bottleneck that prevents scaling to new problems (Fromer and Coley, 2024; Bharti et al., 2024).

Recently, Large Language Models (LLMs) have emerged as scientific agents (Ren et al., 2025) capable of automating design workflows (Zhong et al., 2024; Lv et al., 2025). This paradigm shows potential for scaling up the manual, case-by-case tuning of simulator parameters. However, LLMs often lack specific priors (e.g., cost-awareness), particularly for niche or specialized simulators, leading to potential performance degradation. To remedy this lack of prior knowledge, these agentic systems typically rely on **inference-time scaling**—generating and evaluating multiple reasoning paths to select the best outcome (Roohani et al., 2025; Liu et al., 2024). While effective for increasing

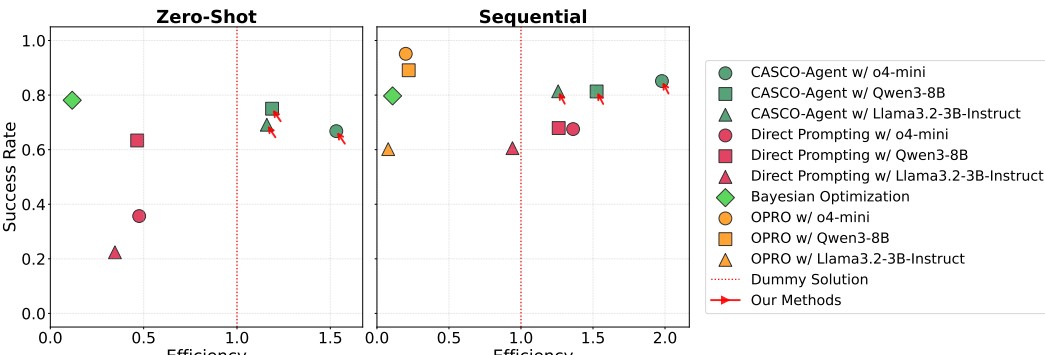

Figure 1: Performance on *Heat 1D* , *Euler 1D* , and *NS Transient 2D* simulators (Single-Turn left, Multi-Turn right). *Efficiency* (↑) denotes normalized cost of successful runs; *Success Rate* (↑) denotes the ratio of valid simulations. *CASCO-Agent* (Ours) achieves Pareto dominance over OPRO (Yang et al., 2023) and BO (Nogueira, 2014) across all base models.

exploration width, this strategy is disastrous for physics simulations: "verifying" every path requires querying the expensive simulator, rendering standard inference scaling prohibitively costly.

To mitigate this, some approaches employ neural networks (NN) as proxies for the simulator (Lyu et al., 2024). However, two main issues persist: 1) standard surrogates attempt to approximate complete high-dimensional physical fields, making them data-hungry and prone to overfitting specific conditions; and 2) they lack explicit cost-awareness. To address these challenges, we propose **C**ost-**A**ware **S**imulation **C**onfiguration **O**ptimization Agent (*CASCO-Agent*): a framework that shifts the surrogate target from high-dimensional physical fields to low-dimensional metrics (cost and accuracy). This strategy simplifies training to scalar targets and improves generalization, as cost dynamics are driven by universal parameters (e.g., mesh size, time integration) rather than specific physical conditions. Crucially, these lightweight surrogates are computationally negligible compared to the simulator, enabling *CASCO-Agent* to perform massive, parallel inference-time scaling to optimize exploration without incurring prohibitive costs.

Our contributions are summarized as follows:

1. We introduce *CASCO-Agent*, a framework integrating inference-time scaling with lightweight cost-efficiency surrogates. To our knowledge, this is the first method to explicitly consider efficiency along side from accuracy in LLM automating physics based simulations.

2. We demonstrate across three diverse PDE solvers (elliptic, parabolic, hyperbolic) that *CASCO-Agent* significantly outperforms Bayesian optimization and LLM baselines, achieving Pareto-optimal efficiency.

3. We release a comprehensive benchmark for cost-aware physics simulation design, including open-source environments and evaluation protocols.

## 2 RELATED WORK

**Black-Box Optimization & Benchmarks.** Bayesian optimization (BO) is an universal approach for tuning experimental parameters (Snoek et al., 2012; Knudsen et al., 2021), with recent advances integrating pre-trained surrogates to improve initial sampling (Wang et al., 2024a; Fan et al., 2022). There also exists BO benchmarks like Design-Bench (Trabucco et al., 2022) and Inverse-Bench (Zheng et al., 2025a) cover scientific problems, they rarely treat *computational cost* as part of the optimization target. This limits their utility to mature simulators where configurations are already established and the cost can be simplified as simulation counts. We address this gap by explicitly augmenting the usual accuracy metric with computational cost in our dataset. We also incorporate BO methods, both general and cost-aware versions (which penalize the sampling probability of higher-cost runs) (Gorecki et al., 2023; Bharti et al., 2024)as our baselines.

**Scientific Agents & Related Benchmarks.** LLMs have been applied to autonomous experiment design (Boiko et al., 2023; Lu et al., 2024) and hypothesis generation (Wang et al., 2024b; Zheng

et al., 2025b). However, these are largely feasibility studies; they demonstrate *capability* rather than *efficiency*. The usual high pass@k (e.g., k=1024) metrics in these works often hide the massive computational cost of failed trials. In this regard, MLEBench (Chan et al., 2025), which tracks training costs for ML tasks, is similar to our work in considering tool costs. However, no equivalent benchmark exists for LLM agents in physics-based simulations. Our work brings this necessary attention to the community. We also incorporate a straightforward agentic workflowdirect promptingas a baseline.

**LLMs as Optimizers.** Approaches like OPRO (Yang et al., 2023) and LLM-assisted evolutionary algorithms (Hao et al., 2024) use LLMs as iterative optimizers. While promising, they create massive parallel paths, each requiring a simulator query, thereby incurring prohibitive costs. There have been very few attempts to include cost considerations in LLM agents (Song et al., 2024; Wu et al., 2024); however, these typically assume fixed tool costs (e.g., using function-as-a-service), failing to capture the relationship between cost and configuration parameters (e.g., mesh node number, time integration size). *CASCO-Agent* is the first work to systematically include cost considerations while solving the rigid cost assumption by modeling cost as FLOPs complexity, thus accurately capturing the relationship between cost and configurations.

## 3 METHODOLOGY

### 3.1 PROBLEM DEFINITION

Given design variable space $\mathcal{X}$ (e.g., spatial/temporal resolution, spatial interpolation methods), environmental parameter space $\Theta$ (e.g., initial or boundary conditions), and output observation space $\mathcal{Y}$, we define the forward simulation-based experimental process as $\mathcal{F} : \mathcal{X} \times \Theta \to \mathcal{Y}$:

$$y = \mathcal{F}(x, \theta), \quad \text{where } x \in \mathcal{X}, \, \theta \in \Theta \tag{1}$$

With utility function $\Phi : \mathcal{Y} \times \Theta \to \mathbb{R}$ (e.g. representing accuracy or physical validity of simulated results) and cost function $\mathbf{C} : \mathcal{X} \times \mathcal{Y} \times \Theta \to \mathbb{R}$ (e.g. wall time, complexity analysis, RAM consumption), the *CASCO* problem becomes:

$$x^* = \arg\max_{x \in \mathcal{X}} \left( \Phi(y, \theta), \, -\mathbf{C}(x, y, \theta) \right) \tag{2}$$

In this work, we define computational cost as the number of floating point operations (consistent with complexity analysis) and normalize cost relative to a brute-force reference (dummy) solution $z_\theta$ that satisfies accuracy requirements with optimal cost (within a coarse search granularity):

$$\hat{\mathbf{C}}(x, y, \theta) = \frac{\mathbf{C}(x, y, \theta)}{\mathbf{C}(z_\theta, \theta)}. \tag{3}$$

Following previous works (Snoek et al., 2012; Fromer and Coley, 2024), we combine the normalized cost and utility objectives into a single reward metric for an experiment $(x, y, \theta)$:

$$\mathcal{R}^0(x, y, \theta) = \frac{\Phi(y, \theta)}{\hat{\mathbf{C}}(x, y, \theta)} \tag{4}$$

We consider two variants of the *CASCO* problem: Single-Turn *CASCO*, where the algorithm proposes only one configuration, and Multi-Turn *CASCO*, where the algorithm proposes a trajectory of configurations for iterative refinement (Huan et al., 2024; Bharti et al., 2024)):

**Definition 3.1** (Single-Turn **C**ost-**A**ware **S**imulation-Based **C**onfiguration **O**ptimization (*CASCO*) )**.**

$$\mathcal{Q}_0 : x^* = \arg\max_{x \in \mathcal{X}} \mathcal{R}^0(x, y, \theta), \tag{5}$$

**Definition 3.2** (Multi-Turn **C**ost-**A**ware **S**imulation-Based **C**onfiguration **O**ptimization (*CASCO*) )**.**

$$\mathcal{Q}_m : \{x\}^* = \arg\max_{\{x_1, \ldots, x_n\} \in \mathcal{X}^*} \mathcal{R}^s(\{x_1, \ldots, x_n\}, \{y_1, \ldots, y_n\}, \theta), \tag{6}$$

where $\mathcal{X}^*$ is a sequence consisting of an arbitrary number of elements from $\mathcal{X}$. In this work, we allow multi-turn solutions with any length. $\{y\} = \{y_1, y_2, ..., y_n\}$ are observations from sequence $\{x\} = \{x_1, x_2, ..., x_n\}$, and the modified multi-turn reward $\mathcal{R}^m$ is:

$$\mathcal{R}^m(\{x_1, \ldots, x_n\}, \{y_1, \ldots, y_n\}, \theta) = \frac{\max_i \Phi(y_i, \theta)}{\sum_i \hat{\mathbf{C}}(x_i, y_i, \theta)}, \tag{7}$$

i.e., the ratio between maximum utility and total cost incurred by this sequence of proposals.

The two variants of the *CASCO* problem, $\mathcal{Q}_0$ and $\mathcal{Q}_m$, are distinct and have different metrics with different reference solution $z_\theta$. They evaluate different abilities of the solution: $\mathcal{Q}_0$ requires an intuitive choice of simulation parameter, while $\mathcal{Q}_m$ requires adaptation based on simulation feedback. They are not to be recognized as the same task with a varying hyperparameter (number of turns).

## 3.2 COST-AWARE SIMULATION CONFIGURATION OPTIMIZATION AGENT

**Overview.** We adopt the inference-time scaling framework of Optimization by PROmpting (OPRO) (Yang et al., 2023; Song et al., 2024; Chen et al., 2022), with the addition of a module that efficiently provides utility $\Phi(x, y, \theta)$ and cost $\hat{\mathbf{C}}(x, y, \theta)$ information without calling the expensive ground-truth simulations. Specifically, we train a neural-network surrogate to predict these scalar signals from the design variables and environmental parameters. Because the scalar outputs are strongly correlated with a few key design variables, signal model training converges with fewer samples and smaller model size, compared with full-physics surrogates (Ghafariasl et al., 2024; Hou and Evins, 2024); see C for details. The signal model then supplies feedback, the predicted utility and cost, to the LLMs proposed parameter designs. These feedback signals, recorded as designvalue pairs, are appended to the prompts history as in-context examples to aid the LLMs optimization output. See Figure 2 for an illustration of *CASCO-Agent*s workflow.

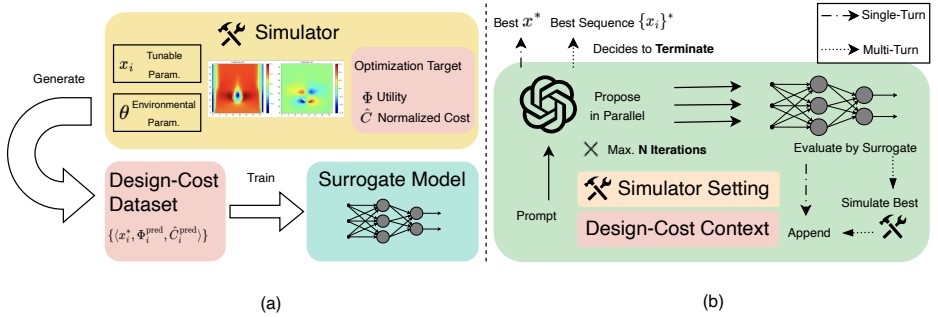

(a)                                                                 (b)

Figure 2: Overview of *CASCO-Agent*. **(a)** For a given simulation-based experimental design problem, *CASCO-Agent* samples uniformly within the design space to train a neural network surrogate for feedback signals of utility and cost. **(b)** At inference time, the LLM agent is prompted with explanations of the simulator's setting and together with in-context examples of designvalue pairs. It then proposes an ensemble of candidate designs in parallel. The LLM queries the surrogate model for feedback and augments the context with the new designscore pairs for the next round of proposal generation (Single-Turn Setting), or it evaluates the surrogate-selected best candidate with the actual simulator to obtain ground-truth feedback (Multi-Turn Setting). The agent outputs the best design at the final iteration or terminates early when a satisfactory and stable solution has been reached.

**Signal Neural Network.** We train lightweight networks $\mathcal{S} : \mathcal{X} \times \Theta \to \mathbf{D}_\Phi \times \mathbb{R}$ to predict utility and cost signals only, where $\mathbf{D}_\Phi$ is the short-hand for the range of utility function $\Phi$. Our experiments show that small fully connected neural networks can learn the function well for the experiments in this paper, though we note that architecture and model size can be adapted according to the need of specific solvers. See Appendix C for details on neural network implementation for this paper.

To provide rich, informative utility signals, as opposed to the binary boolean signals in prior works (Smucker et al., 2018; Huan et al., 2024), we design a reward shaping function $f$ that maps the binary experiment outcome $b(y) \in \{0, 1\}$ to a scalar soft success measure $f(y) \in [0, 1]$ defined as follows.

**Definition 3.3** (Soft Utility Function). Following the notations in section 3.1, let $\mathcal{Y}$ be the experiment's observation space, $\Theta$ be the environmental variable space, and $\Phi$ be the original utility function. Define the feasible set

$$\mathcal{G}_\theta := \{\, y \in \mathcal{Y} : \Phi(y, \theta) = 1 \,\}. \tag{8}$$

We call a mapping $f : \mathcal{Y} \times \Theta \to [0, 1]$ a *soft utility function* if it satisfies:

(i) Feasibility calibration: $\forall y \in \mathcal{G}_\theta : \ f(y) = 1, \quad \sup_{y \notin \mathcal{G}_\theta} f(y) \ < \ 1.$

(ii) Normalization: $0 \ \le \ f(y) \ \le \ 1, \quad \forall y \in \mathcal{Y}.$

(iii) Monotone alignment: $\Phi(y_1) \ \preceq \ \Phi(y_2) \implies f(y_1) \ \le \ f(y_2).$

The signal neural network $\mathcal{S}$ learns the soft utility signal $f(y)$ in the place of $\Phi(y)$. We provide the following proposition that any soft utility function $f$ guarantees an incremental performance over binary utility functions when integrated into our framework, and a well-designed $f$ will lead to more significant improvements. Refer to Appendix D for our design of $f$ and proofs of the proposition.

**Definition 3.4** (Policies). Recall that $R^0(x, y, \theta)$ and $\mathcal{R}^m(\{x_1, \ldots, x_n\}, \{y_1, \ldots, y_n\}, \theta)$ are respectively single-turn and multi-turn reward defined in Eq.1). For a task instance $\theta$, we define two policies:

(i) **Binary-utility policy** $\pi_{\mathrm{bin}}(x_t \mid \theta, h_{t-1})$: at step $t$, given history $h_{t-1} = \{(x_s, y_s, b(y_s, \theta))\}_{s=1}^{t-1}$, sample the next design $x_t$; denote the induced distribution over the final design by $x \sim \pi_{\mathrm{bin}}(\cdot \mid \theta)$.

(ii) **Soft-utility policy** $\pi_f(x_t \mid \theta, h_{t-1})$: replace $b$ with any soft utility $f$ from Definition 3.3, i.e., the history stores $(x_s, y_s, f(y_s, \theta))$. Denote the resulting final-design distribution by $x \sim \pi_f(\cdot \mid \theta)$.

**Proposition 3.5** (Soft utility dominates binary utility in expected reward). *Fix a base model and any soft utility $f$ in Definition 3.3, the expected reward under the soft-utility policy is no worse than under the binary-utility policy:*

$$\mathbb{E}_\theta \, \mathbb{E}_{x \sim \pi_f^0(\cdot \mid \theta)} \big[\, R^0(x, \theta) \,\big] \ \ge \ \mathbb{E}_\theta \, \mathbb{E}_{x \sim \pi_{\mathrm{bin}}^0(\cdot \mid \theta)} \big[\, R^0(x, \theta) \,\big].$$

$$\mathbb{E}_\theta \, \mathbb{E}_{\{x\} \sim \pi_f^m(\cdot \mid \theta)} [R^m(\{x\}, \theta)] \ \ge \ \mathbb{E}_\theta \, \mathbb{E}_{\{x\} \sim \pi_{\mathrm{bin}}^m(\cdot \mid \theta)} [R^m(\{x\}, \theta)].$$

In summary, for a given simulation-based experimental design task, we train a lightweight network $\mathcal{S} : \mathcal{X} \times \Theta \to \mathbf{D}_\Phi \times \mathbb{R}$ to predict a certain design's utility and cost; in cases where utility function $\Phi$ is sparse and less informative, we substitute it with soft utility function $f$ and learn soft utility signals, i.e. we learn $\mathcal{S} : \mathcal{X} \times \Theta \to \mathbf{D}_y \times \hat{\mathbf{C}}$, where $\mathbf{D}_y$ is the range of $f$. The trained network $\mathcal{S}$ provides feedback for the following agent's self-refinement.

**Agentic Framework.** The agent leverages Optimization by PROmpting (OPRO) as the base LLM in-context optimization method, and use the signal network's feedback as in-context examples. We note that expanding to other inference-time scaling methods is straightforward and requires no change or re-training of the signal neural network. Pseudocode for our agent implementation is provided in Appendix B.

For Single-Turn *CASCO*, the agent starts with 5 (a hyper-parameter to adjust based on inference budget) uniformly-sampled tuples of (design variable, utility, efficiency) evaluated by surrogate neural network. Then the agent iteratively proposes ensembles of candidate design choices, receives neural network feedback for the entire ensemble, and append them to the example pool. The example pool is managed as a priority queue with key (utility, efficiency) and presented to the model in ascending order.The example pool only keeps top-10 samples (also a hyper-parameter) to concise the context. The process is repeated for a fixed number of iterations, and the best design in the example pool is chosen for the final design. The fixed number of iterations is another hyperparameter reflecting the allowed LLM inference budget.

To solve Multi-Turn *CASCO*, we warm-start with Single-Turn *CASCO* solution for the first round of ground truth simulator evaluation, and then append the results to the pool. This process is repeated for each iteration to find the most promising proposals for simulator evaluation. In short, the Single-Turn *CASCO* works as an acquisition function for each of the multi-turn steps. The loop terminates when either the LLM decides that a satisfactory solution is found or the computation cap is reached.

## 4 EXPERIMENTS

### 4.1 EXPERIMENTAL ENVIRONMENT

We demonstrate the ability of *CASCO-Agent* on three physics simulators: (1) 1D heat conduction equation with mixed boundary conditions, (2) 1D compressible inviscid flow with Euler equation, and (3) 2D transient incompressible Navier–Stokes equation, referred to as *Heat 1D* , *Euler 1D* and *NS Transient 2D* respectively, for brevity. Appendix A contains details on the design variable space $\mathcal{X}$, observation space $\mathcal{Y}$, and parameter space $\Theta$. We focus on spatial resolution tuning tasks, where the tunable parameter governs the spatial resolution of the simulation, creating a trade-off between simulation accuracy and computational cost. The tunable parameters in our experiments are:

1. The number of grid numbers ($n\_space$) for *Heat 1D* and *Euler 1D*
2. The grid resolution along X-axis ($resolution$) for *NS Transient 2D*

We design three precision level goals $\delta$ for each task, reflecting moderate to stringent accuracy requirements in real-world experiments. For each task and each precision level, we evaluate the methods on around 25 settings varying in environmental parameters.

For each problem instance characterized by $\theta$, we first obtain a (near-)optimal design $z_\theta$ via brute-force search that guarantees successful convergence, e.g. through iteratively doubling the parameter until successful, serving as a reference point for both accuracy and cost. This is solely for the evaluation of our method and not necessary in practice. We then define the success of the simulation through the following utility function:

$$\Phi(\mathcal{F}(x,\theta),\theta) = \mathbf{1}\{\ ||\mathcal{F}(x,\theta) - \mathcal{F}(z_\theta,\theta)||_2 \ \leq \ \delta\ \},\qquad(9)$$

Where $\mathbf{1}$ is the indicator function, $||\cdot||_2$ is the root mean square error across dimensions of the observation space, and $\delta$ is a tolerance parameter reflecting various precision needs in real-world applications. The success rate is defined as the ratio of successful simulations where $\Phi(\mathcal{F}(x,\theta)) = 1$. The cost $\mathbf{C}$ is defined as previously introduced in our problem formulation.

### 4.2 BASELINES AND SETTING

We compare our results against the following baselines. **Bayesian Optimization** (*BO*): We use a classic implementation (Nogueira, 2014) with Gaussian Process (GP) (Rasmussen, 2004) and Upper Confidence Bound (UCB) (Berk et al., 2020). We used consistent training samples for the signal neural network *CASCO-Agent* and for the GP regressor to achieve a fair comparison. **Direct query to LLM medels** (*Direct Query*) and the original **Optimization by Prompting** (*OPRO*) (Yang et al., 2023) are LLM-based approaches. For all LLM-based methods (including our *CASCO-Agent*), we design a shared set of prompts explaining the Physics scenario, optimization target and simulator calling APIs; refer to E for examples. Notably, *OPRO* requires repeated evaluations of the ground-truth simulator; therefore, we restrict its use to the Multi-Turn setting.

For the implementation of *CASCO-Agent*, we trained a lightweight neural network for each task (*Heat 1D* , *Euler 1D* , and *NS Transient 2D* ) separately, each with approximately 10k parameters and trained on about 4k sampled points per problem. The networks outputs are the RMSE to the reference solution and the cost. At inference time, we map the predicted RMSE to a utility signal using the soft utility functions described in Definition 3.3; we also compare using the binary utility $\Phi$ in ablation studies. See Appendix C for details.

### 4.3 METRICS

For ease of future reference, we denote the optimization targets in 5 and 6 as respectively $R^0$ and $R^m$, referring to them as Single-Turn or Multi-Turn *Reward Functions*. We also report success rates

$P^0$ and $P^m$ to help us better understand the qualities of proposed solutions.

$$R^0 = \frac{\Phi(\mathcal{F}(x,\theta),\theta)}{\hat{\mathbf{C}}(x,\theta)}, \quad R^m = \frac{\max_i \Phi(\mathcal{F}(x_i,\theta),\theta)}{\sum_i \hat{\mathbf{C}}(x_i,\theta)}$$

## 4.4 ANALYSIS

We refer readers to Table 3 of F for complete results in three scenarios; here we report the following findings that help understand and verify the efficacy of our method.

**Our method outperforms most baselines in terms of $R^0$ and $R^m$.** As shown in Figure 3 and Figure 1, our method outperforms all comparisons in the Single-Turn setting and all but a few exceptions in the Multi-Turn setting. We argue that these suboptimal cases are due to the inferior reasoning ability of open-source models, causing them to occasionally fail to refine their solutions based on feedback. Note that in many cases, especially in the easier scenarios *Heat 1D* and *Euler 1D*, OPRO and BO are significantly worse than Direct Query, whereas our method is significantly better. This is because convergence is relatively easy in such scenarios, so the additional ground-truth simulator calls used by OPRO and BO incur extra cost without meaningfully improving the solution. Our method does not require additional ground-truth simulator queries.

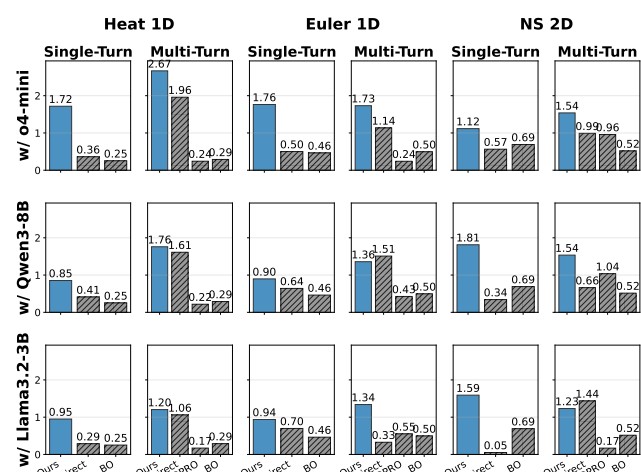

Figure 3: **Comparison of Single-Turn and Multi-Turn rewards for all methods.** Each bar shows the mean reward, averaged over all precision levels of a task, for methods on a given base model. As discussed in definition 4.2, OPRO is only considered in the Multi-Turn scenarios. BO is plotted alongside LLM methods for clarity of comparison.

**Our method delivers substantial reward gains over Direct Query, especially on medium- and easy-difficulty tasks; on harder tasks, it consistently improves success rate.** As shown in Figure 4, reward improvements are most pronounced in easier scenarios (*Heat 1D*; low-precision *Euler 1D*). In harder scenarios (medium- to high-precision *Euler 1D*; *NS Transient 2D*), reward gains are smaller, but success rate improves steadily. This pattern suggests an intrinsic optimization behavior: for unfamiliar questions, *CASCO-Agent* first optimizes correctness, and then optimizes efficiency.

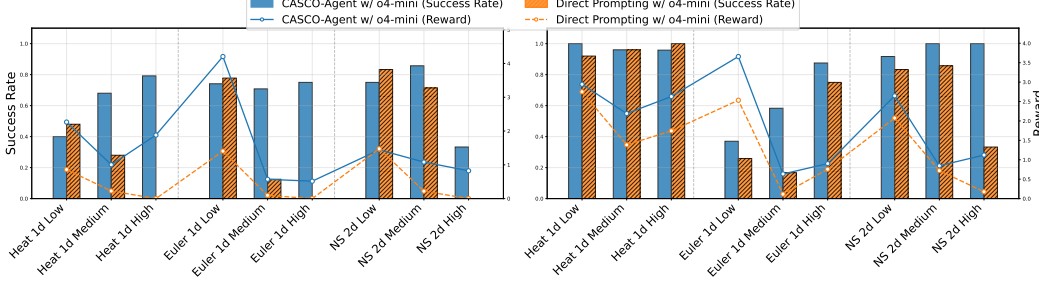

Figure 4: **Reward ($R^0$, $R^m$) and success rate ($P^0$, $P^m$) across all difficulty levels** in the **Single-Turn setting (left)** and the **Multi-Turn setting (right)**. Tasks are ordered by increasing difficulty: *Heat 1D*, *Euler 1D*, *NS Transient 2D*. Our methods improvements in reward are largest on easy-to-medium tasks and remain present on hard tasks.

## 4.5 ABLATIONS

We present ablation studies for the two main components of *CASCO-Agent*: the surrogate neural network and the LLM agent. All ablation studies are preformed on the same set of problems, *Euler 1D* with medium precision level, with base model OpenAI o4-mini (OpenAI et al., 2024).

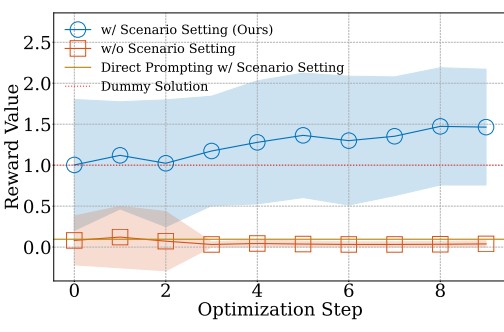

Figure 5: Mean Reward over optimization Steps for *CASCO-Agent*, with or without scenario setting description in prompt.

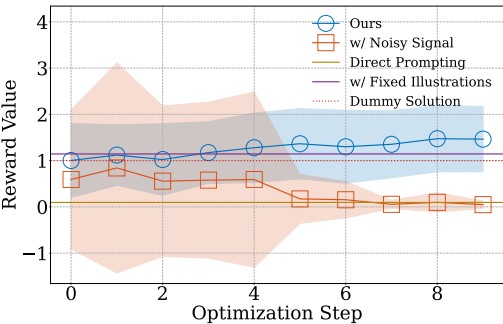

Figure 6: Mean Reward over optimization Steps for *CASCO-Agent*, either with surrogate signal, noisy signal or fixed illustrations.

**Physics prior knowledge is necessary to achieve in-context optimization in our tasks.** Figure 5 presents an ablation study on whether the scenario setting is included in the LLM's prompt. We argue that the merits of utilizing LLM in our framework lie in both their in-context optimization abilities and their prior domain knowledge. For the alternative setting (orange lines in 5), we only prompt the model to solve the problem as a numerical optimization problem; see the prompts in E. Figure 5 shows that *CASCO-Agent* (blue lines), with physics prior knowledge, can consistently improve reward to surpass baselines, whereas the trajectory without scenario description fails to achieve improvements and converges to a low-reward local optimum. This behavior is also visible in a case study illustrated in 11a.

**Feedback signals are important for agent optimization in our tasks.** We study the effects of our surrogate signal network and present the results in figure 6. We compare *CASCO-Agent* with (1) in-context optimization with a fixed set of ground-truth examples for all problems, and (2) our agent equipped with noisy signal from a poorly fitted surrogate model. We experiment on both Single-Turn and Multi-Turn settings in *Euler 1D* 's medium precision level with base model GPT-4o-mini. As shown in Figure 6 and 11b, in both the dataset-level pattern and the case study, our method starts from a worse point than that of fixed illustrations', but surpasses it in later optimization steps; the noisy signal fails to guide the model's optimization after the first few steps, highlighting the importance of an effective signal model.

**Soft surrogate signals significantly improve optimization performance compared to binary surrogate signals.** We verify the effectiveness of the soft utility (Definition 3.3). Specifically, we compare Single-Turn results of our framework under two variants: (a) integrating surrogates with the original binary utility function, and (b) our approach that uses a soft utility function in the surrogate signals. As shown in Figure 7, the soft-utility variant achieves significantly better performance at the dataset level and exhibits a steadier upward trend in the case study.

We also present a case study in 7b, which plots the predicted reward **(dashed lines)** of the step-wise optimal design for both methods besides the real reward in **solid lines**. As shown by the orange dashed line in 7b, once the model receives a zero-utility signal from the surrogate at step 3, it stops refining and remains at a local optimum. By contrast, the blue line shows that although the model proposes the same point at step 3, the non-zero soft-utility signal it receives enables it to continue refining the solution.

Complete ablation study results are presented in Table 4 of Appendix F. We show that each component of *CASCO-Agent*, including the physics prior, the signal NN, and the prompt design all contribute to the final performance. *CASCO-Agent* achieves the Pareto optima of success rate and efficiency for all settings, as shown in Figure 1.

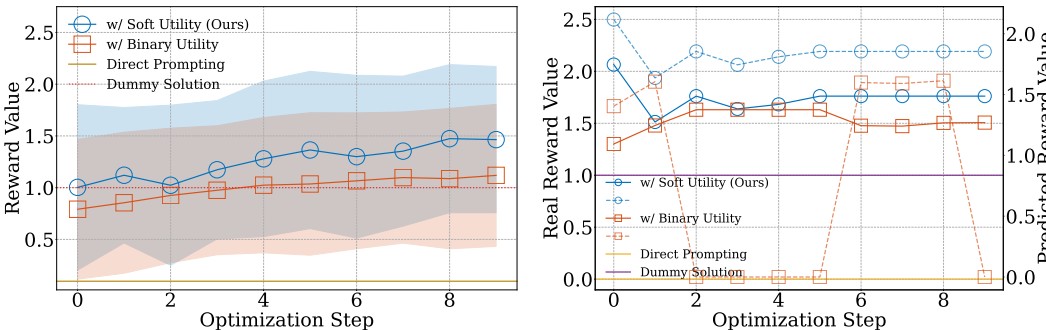

(a) Mean Reward over Optimization Steps for *CASCO-Agent*, using different functions for surrogate signal.

(b) Case study. An exemplar optimization trajectory in Single-Turn setting. Notations explained in 4.5.

Figure 7: **Study on soft utility functions vs. binary signals for surrogate signal** for o4-mini.

**Surrogate Flexibility.** Our framework is agnostic to the underlying surrogate model. While neural networks excel with abundant data, they struggle in data-scarce regimes ($< 500$ samples, see Table 1). To address this, we demonstrate that the neural backbone in *CASCO-Agent* can be seamlessly substituted with a Gaussian Process (GP) regressor (Brochu et al., 2010; Shahriari et al., 2016), utilizing its posterior mean and variance as feedback signals. This allows *CASCO-Agent* to match strong BO baselines even when data is limited. Since the LLM optimizes based on open-form numerical feedback rather than internal model states, users can adapt the surrogate (NN, GP, or others) to match their specific data availability.

Despite this flexibility, we suppose that the data-abundant regime, and thus the choice of neural networks, is the most practical default. The primary objective of *CASCO-Agent* is to automate parameter tuning in labs previously reliant on manual workflows. Such labs naturally possess extensive archives of historical simulation logs. By simply mining these logs (e.g., via file timestamps), we can construct the large datasets required for training.

Table 1: Performance comparison between *CASCO-Agent* with NN, *CASCO-Agent* with GP and Bayesian Optimization under varying sample sizes. The GP regressor integrated into *CASCO-Agent* is trained with exactly the same hyper-parameters and dataset as the one used for BO.

|  | 50 samples | 100 samples | 500 samples | full samples (~4k) |
|---|---|---|---|---|
| *CASCO-Agent* with NN | 0.142 | 0.239 | **0.471** | **0.571** |
| BO | 0.459 | **0.418** | 0.392 | 0.391 |
| *CASCO-Agent* with GP | **0.465** | 0.354 | 0.205 | 0.342 |

## 5 CONCLUSION

We presented the **C**ost-**A**ware **S**imulation **C**onfiguration **O**ptimization Agent, a LLM Agent framework for experimental design that focuses on cost-efficiency. Through experiments on three physics simulator environments, each with varying environmental setting and precision requirements, we demonstrated that *CASCO-Agent* consistently outperforms both classical Bayesian optimization baselines and state-of-the-art LLM-based optimizers. Our results highlight its ability to achieve high success rates and favorable cost-efficiency trade-offs, even when direct evaluations are prohibitively expensive. Our method introduces the novel contribution of utilizing a low-dimensional cost-efficiency signal neural network, which through our ablation studies we show significantly improves utility of both single-turn and multi-turn experiment design. These findings suggest that *CASCO-Agent* provides a practical and scalable path toward deploying agentic frameworks in experiment design in scientific discovery pipelines.

Our approach has the following limitations to be explored in future work. The accuracy of *CASCO-Agent* depends on the fidelity of the surrogate, which may under-fit in highly complex or noisy experimental landscapes, and requires some degree of human tuning. Moreover, our data sampling strategy does not guarantee the minimization of sampling size while the model converges. Future work can aim to address these limitations by exploring richer surrogate models, adaptive sampling strategies,

and tighter coupling between surrogate predictions and target function evaluation to improve the quality of feedback to LLM. Extending *CASCO-Agent* to multi-objective, higher-dimensional, or real-world experimental systems will further test its scalability and practical utility, paving the way toward more autonomous and cost-efficient experimental design agents.

## REPRODUCIBILITY

We evaluate this work on three physics-solver environments that we implemented: *Heat 1D* , *Euler 1D* , and *NS Transient 2D* which include solvers, reference solutions, problem sets, and evaluation pipelines. We plan to extend and organize these into a benchmark to aid the open-source community in solving **C**ost-**A**ware **S**imulation-Based **C**onfiguration **O**ptimization (*CASCO*) better. As the benchmark is still in progress, our solvers, evaluation pipeline, etc. may not yet be robust enough for convenient reproduction. Therefore, we consider it appropriate to open-source the code for this work after acceptance, including not only a (subset) of the aforementioned benchmark but also the neural-network training, the main framework, and the plotting components.

## ETHICS STATEMENT

This work studies cost-aware experimental design agents for physics simulations (e.g., 1D Heat Conduction and Euler equations) and does not involve human subjects, personal data, or sensitive attributes. All data are synthetic or standard simulation benchmarks; no personally identifiable information is used or created. We comply with licenses and usage terms for third-party software and models; any proprietary APIs were accessed under their respective terms.

Potential risks are limited. As our method can improve search efficiency, there is a generic risk of misuse to optimize unsafe physical systems. To mitigate this, we focus on pedagogical and widely used benchmark scenarios with explicit constraints and provide documentation intended for scientific replication rather than domain-specific exploitation.

Fairness and demographic bias considerations are not applicable to our setting. The environmental impact is modest: we train lightweight surrogates on small datasets and use limited inference budgets; we report hardware and runtime details to enable carbon accounting. For reproducibility, we will release code, configurations, and seeds, and follow standard reporting checklists. We declare no conflicts of interest and no concurrent submissions related to this work.

## THE USE OF LARGE LANGUAGE MODELS

In this work, Large Language Models are primarily used for assisting in polishing the mathematical formulation in 3.3, explaining the results in 4.5 and generating the plotting code for Figure 1, 3, 4, 5, 6 and 7.

They are also used for polishing text in some sections. They were NOT used in research ideation and/or writing.

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

## A Experimental Environment

**Heat Transfer 1D. (*Heat 1D* )**    This solver addresses the 1D heat conduction equation:

$$\frac{\partial T}{\partial t} = \alpha \frac{\partial^2 T}{\partial x^2}$$

using explicit finite difference methods with natural convection boundary conditions at $x = 0$ and adiabatic conditions at $x = L$. The tunable parameters include the spatial resolution (`n_space`) and the CFL number (`cfl`) that determines the simulation time step by:

$$\Delta t = \texttt{cfl} \times \frac{(\Delta x)^2}{2\alpha},$$

where $\alpha$ is the thermal diffusivity. The computational cost follows the relationship $C = \texttt{n\_space} \times \texttt{n\_t}$, where `n_t` is the number of time steps accumulated in the solver. The metric for convergence is the RMSE of the heat flux at the convection boundary at the final time step. This simulation has 25 different profiles with varying initial uniform temperatures and physical properties, generating 148 tasks in total, counting both Single-Turn and Multi-Turn settings.

**Euler 1D. (*Euler 1D* )**    This solver implements the 1D Euler equations for compressible flow:

$$\frac{\partial \mathbf{U}}{\partial t} + \frac{\partial \mathbf{F}(\mathbf{U})}{\partial x} = 0$$

using the MUSCL-Roe method with superbee limiter for high-resolution shock capturing. The tunable parameters include the CFL number (`cfl`) that determines the simulation time step by:

$$\Delta t = \texttt{cfl} \times \frac{\Delta x}{|\lambda|_{\max}},$$

where $|\lambda|_{\max}$ is the maximum eigenvalue of the flux Jacobian, the spatial resolution (`n_space`), the limiter parameter `beta` for generalized minmod flux limiter, and the blending parameter `k` between 0-th and 1-st order interpolation scheme. The computational cost follows the relationship $C = \texttt{n\_space} \times \texttt{n\_t}$, where `n_t` is the number of time steps accumulated in the solver. Convergence is evaluated through multiple criteria: RMSE of the solution fields, positivity preservation of density and pressure, and shock consistency validation. The dataset encompasses 3 classical benchmark profiles (Sod shock tube, Lax problem, and Mach 3), generating a total of 134 tasks, counting both Single-Turn and Multi-Turn settings.

**Transient Navier-Stokes 2D. (*NS Transient 2D* )**    This solver implements the 2D transient incompressible Navier-Stokes equations:

$$\frac{\partial u}{\partial x} + \frac{\partial v}{\partial y} = 0$$

$$\frac{\partial u}{\partial t} + u\frac{\partial u}{\partial x} + v\frac{\partial u}{\partial y} = -\frac{\partial p}{\partial x} + \frac{1}{Re}\left(\frac{\partial^2 u}{\partial x^2} + \frac{\partial^2 u}{\partial y^2}\right)$$

$$\frac{\partial v}{\partial t} + u\frac{\partial v}{\partial x} + v\frac{\partial v}{\partial y} = -\frac{\partial p}{\partial y} + \frac{1}{Re}\left(\frac{\partial^2 v}{\partial x^2} + \frac{\partial^2 v}{\partial y^2}\right)$$

where $u, v$ are velocity components, $p$ is pressure, and $Re$ is the Reynolds number. The tunable parameters include the spatial resolution (`resolution`) that determines the computational grid size, the CFL number (`cfl`) controlling time step stability through $\Delta t = \texttt{cfl} \times \Delta x$, the relaxation factor (`relaxation_factor`) for pressure correction convergence, and the residual threshold (`residual_threshold`) for pressure solver convergence. The computational cost follows the relationship $C = 2 \times \texttt{resolution}^2 \times (\texttt{num\_steps} + \texttt{total\_pressure\_iterations})$, where the factor of 2 accounts for the fixed aspect ratio domain configuration with $x\_resolution = 2 \times \texttt{resolution}$. Convergence is evaluated through normalized velocity RMSE criteria, with temporal evolution tracked throughout the simulation. The dataset encompasses 18 benchmark profiles across 6 different boundary conditions (simple circular obstacles, complex geometries, random obstacle fields, dual inlet/outlet configurations, dense obstacle arrays, and dragon-shaped obstacles) tested at three Reynolds numbers (Re=1000, 3000, 6000), generating a total of 44 tasks across different precision levels and geometric complexities.

**Dummy Solution Search.** For each task, we find optimal solutions that meet both accuracy requirements and have the lowest cost using brute-force search. Given our parameters have a monotonic relationship between cost and accuracy (i.e., they are spatial resolution), we start with a coarse value and Multi-Turnly refine it with fixed ratios (e.g., halve the time step size, double the spatial resolution) until the distance between adjacent runs is within the accuracy threshold. For single-turn tasks, we set the reference cost to the optimal cost found by brute-force search. For multi-turn tasks, we set the reference cost to the accumulated cost incurred during the brute-force search.

# B   ALGORITHMIC DIAGRAM

---

**Algorithm 1** *Solve*, Single-Turn *CASCO-Agent* Framework

---

1: **Input:** Forward experimental process $\mathcal{F}$, design space $\mathcal{X}$, environment parameters $\theta$, neural surrogate $\mathcal{S}$, number of iteration $N$, history context length $K$, initial sample size $m$.
2: Initialize LLM design-value history as a priority queue $\mathcal{M}$
3: Push to $\mathcal{M}$ uniformly sampled initial design-value pairs $\{(x_j, \Phi_j^{pred}, \hat{\mathbf{C}}_j^{pred})\}_{j=1}^m$, evaluated by $\mathcal{S}$
4: **repeat**
5:     LLM proposes candidate designs $\{x_i\}_{i=1}^k$
6:     Evaluate candidates with neural surrogate: $(\Phi_i^{pred}, \hat{\mathbf{C}}_i^{pred}) \leftarrow \mathcal{S}(x_i, \theta)$ for $i = 1, \ldots, k$
7:     Push $\{(x_i, \Phi_i^{pred}, \hat{\mathbf{C}}_i^{pred})\}$ to $\mathcal{M}$, keeping only top-$K$ samples.
8: **until** Number of iterations $N$ reached
9: **Output:** $x^* = \arg\max_{x_i} \frac{\Phi(\mathcal{S}(x_i,\theta),\theta)}{\mathbf{C}'(x_i,\theta)}$ from design-value history.

---

**Algorithm 2** Multi-Turn *CASCO-Agent* Framework

---

1: **Input:** Forward experimental process $\mathcal{F}$, design space $\mathcal{X}$, environment parameters $\theta$, neural surrogate $\mathcal{S}$, number of iteration for Single-Turn solution $N$, history context length $K$, initial sample size $m$, maximum allowed number of ground-truth evaluation $T$.
2: Obtain Single-Turn solution $x = Solve(\mathcal{F}, \mathcal{X}, \theta, \mathcal{S}, \mathcal{N}, K, m)$
3: Initialize solution sequence as a queue $\mathcal{A} = \{x_0\}$
4: Initialize LLM ground-truth design-value history as a priority queue $\mathcal{M}$
5: Evaluate with ground-truth simulator $(\Phi_0^{gt}, \hat{\mathbf{C}}_0^{gt}) \leftarrow \mathcal{F}(\S\prime, \theta)$
6: Push $(x_0, \Phi_0^{gt}, \hat{\mathbf{C}}_0^{gt})$ to $\mathcal{M}$
7: **repeat**
8:     LLM agent proposes candidate designs $\{x_i\}_{i=1}^k$
9:     Evaluate candidates with neural surrogate: $(\Phi_i^{pred}, \hat{\mathbf{C}}_i^{pred}) \leftarrow \mathcal{S}(x_i, \theta)$ for $i = 1, \ldots, k$
10:     Add top surrogate-evaluated pair $(x_i, \Phi_i^{pred}, \hat{\mathbf{C}}_i^{pred})$ to solution sequence $\mathcal{A}$
11:     Evaluate with ground-truth simulator $(\Phi_i^{gt}, \hat{\mathbf{C}}_i^{gt}) \leftarrow \mathcal{F}(x_i, \theta)$
12:     Push $\{(x_i, \Phi_i^{gt}, \hat{\mathbf{C}}_i^{gt})\}$ to $\mathcal{M}$, keeping only top-K samples
13: **until** LLM outputs $should\_stop = True$ or number of iterations reaches $T$
14: Outputs $\mathcal{A}$

---

# C   NEURAL NETWORK TRAINING

We train one neural-network for each problem (*Heat 1D*, *Euler 1D*, *NS Transient 2D*)'s all precision levels; each network's input and output dimension are as described in 3.2.

We uniformly sample design and environmental parameters on coarse grids. We specifically include environmental parameters to enable interpolation across conditions while avoiding training and tracking multiple network instances for different environment combinations. We provide the range of inputs (environmental parameters and tunable parameters) as follows, from which we performed uniform sampling, and statistics of sampled targets in Table 2. We stress that while our target dimensions have drastically different ranges and high variance, we perform in-dimension normalization as shown in Figure 9, therefore achieving satisfactory training results shown in Figure 8.

```
Heat 1D:
  Environmental Parameters:
    L: [0.1, 0.3]  # Wall thickness [m] – uniform random in range
    k: [0.5, 1.0]  # Thermal conductivity [W/m-K] – uniform random in
    ↪  range
    h: [0.1, 10000]  # Convection coefficient [W/mš-K] – log-uniform
    ↪  random in range
    rho: [1000, 2000]  # Density [kg/mş] – uniform random in range
    cp: [800, 1000]  # Specific heat [J/kg-K] – uniform random in range
    T_inf: [-40, 40]  # Ambient temperature [řC] – uniform random in
    ↪  range
    T_init: [0, 30]  # Initial temperature [řC] – uniform random in
    ↪  range
    record_dt: 10.0  # Time interval between recordings [s] – fixed
    end_frame: 24  # Simulation end frames – fixed

  Tunable Parameters:
    n_space: [64, 2048]  # Number of spatial points (iterative search:
    ↪  initial=64, factor=2, max_iter=6)

Euler 1D:
  Environmental Parameters:
    L: 1.0  # Domain length – fixed
    gamma: 1.4  # Ratio of specific heats – fixed
    case: {"sod", "lax", "mach_3"}  # Initial condition name – 3
    ↪  discrete values across profiles
    record_dt: {0.02, 0.012, 0.009}  # Time interval between recordings
    ↪  – specific values per case
    end_frame: 10  # Simulation end frames – fixed

  Tunable Parameters:
    n_space: [256, 4096]  # Number of grid cells (iterative search:
    ↪  initial=256, factor=2, max_iter=7)

NS Transient 2D:
  Environmental Parameters:
    boundary_condition: {1, 2, 3, 4, 5, 6}  # 6 boundary condition types
    ↪  across 18 profiles
    reynolds_num: {1000, 3000, 6000}  # Reynolds number – 3 discrete
    ↪  values
    vorticity_confinement: 0.0  # Fixed across profiles
    total_runtime: 1.0  # Fixed across profiles – fixed
    no_dye: False  # Fixed across profiles
    cpu: False  # Fixed across profiles
    visualization: 0  # Fixed across profiles
    advection_scheme: "cip"  # Fixed across profiles

  Tunable Parameters:
    resolution: [50, 400]  # Grid resolution (iterative search:
    ↪  initial=50, factor=2, max_iter=4)
```

Table 2: Dataset Statistics.

|  | $RMSE\ Loss$ | $Cost$ | $N$. samples |
|---|---|---|---|
| *Heat 1D* | $4.47e^{-4} \pm 9.50e^{-4}$ | $8.33e^7 \pm 1.27e^8$ | 4440 |
| *Euler 1D* | $3.48e{-2} \pm 3.60e^{-2}$ | $2.76e^6 \pm 2.42e^6$ | 4020 |
| *NS Transient 2D* | $2.55e^{-1} \pm 1.90e^{-1}$ | $2.11e^8 \pm 1.94e^8$ | 1320 |

For all problems, we train neural-network with the same structure as shown in 9; to achieve optimal results for individual problems, we compare the training results with three sets of structures for

each problem and choose the one with the best test loss. Specifically, we experiment with the combinations of :

```
h: {2, 3, 4, 6}
d: {64, 128, 256}
```

Where $h, d$ follow the notation in 9, and the hyper-parameters we used are shown as follows:

```
activation_mod: ReLU
layer_norm: False
res_connection: False

batch: 16
epochs: 40
steps_per_epoch: 200

peak_lr: 1e-3
weight_decay: 1e-4
warmup_steps: 100
decay_steps: 1000
gnorm_clip: 1.0
accumulation_steps: 100
```

We show the results of our best checkpoints for the three problems in 8.

## D  SOFT UTILITY FUNCTION

*Proof of Proposition 3.5.* Let $b(y, \theta) := \mathbf{1}\{\Phi(y, \theta) = 1\}$ be the binary utility and let $s_f(y, \theta) := f(y, \theta)$ be any soft utility satisfying Definition 3.3. By *normalization* (Def. 3.3(ii)), $f(y, \theta) \in [0, 1]$, and by *feasibility calibration* (Def. 3.3(i)), $f(y, \theta) = 1$ iff $y \in \mathcal{G}_\theta = \{y : \Phi(y, \theta) = 1\}$ and $\sup_{y \notin \mathcal{G}_\theta} f(y, \theta) < 1$. Hence the postprocessing map

$$\tau : [0, 1] \to \{0, 1\}, \qquad \tau(u) := \mathbf{1}\{u = 1\}$$

is well-defined (by normalization) and satisfies $b(y, \theta) = \tau(f(y, \theta))$ for all $(y, \theta)$ (by feasibility calibration). Thus the binary signal is a deterministic garbling of the soft signal.

Fix a base model and budget $T \geq 1$, and write the histories $h_{t-1}^{\text{bin}} = \{(x_s, y_s, b(y_s, \theta))\}_{s=1}^{t-1}$ and $h_{t-1}^f = \{(x_s, y_s, f(y_s, \theta))\}_{s=1}^{t-1}$; then $h_{t-1}^{\text{bin}} = \tau(h_{t-1}^f)$ coordinate-wise. Given any binary-utility policy $\pi_{\text{bin}}$, define a soft-signal policy $\tilde{\pi}_f$ that *simulates* it via

$$\tilde{\pi}_f(\cdot \mid \theta, h_{t-1}^f) := \pi_{\text{bin}}(\cdot \mid \theta, \tau(h_{t-1}^f)).$$

Under identical environment randomness, $\tilde{\pi}_f$ induces the same trajectory distribution as $\pi_{\text{bin}}$, hence

$$\mathbb{E}_{x_T \sim \tilde{\pi}_f(\cdot \mid \theta)}\big[R^0(x_T, \theta)\big] = \mathbb{E}_{x_T \sim \pi_{\text{bin}}(\cdot \mid \theta)}\big[R^0(x_T, \theta)\big] \quad \text{for all } \theta.$$

Taking expectation over the task distribution yields equality in expectation.

By *monotone alignment* (Def. 3.3(iii)), if $\Phi(y_1, \theta) \preceq \Phi(y_2, \theta)$ then $f(y_1, \theta) \leq f(y_2, \theta)$; hence ranking by $f$ is orderpreserving with respect to $\Phi$. Since $R^0(x, \theta)$ (Eq. (4)) is nondecreasing in $\Phi$ (its numerator) and $f = 1$ iff $\Phi = 1$ (by feasibility calibration), using $f$ to refine decisions cannot decrease the expected reward relative to $\tilde{\pi}_f$, and is strictly better whenever such refinements occur with positive probability.

Now let $\pi_f$ denote any soft-signal policy produced by our framework. Since $\pi_f$ can always ignore the extra information and implement $\tilde{\pi}_f$, we have

$$\mathbb{E}_\theta \, \mathbb{E}_{x_T \sim \pi_f(\cdot \mid \theta)}\big[R^0(x_T, \theta)\big] \geq \mathbb{E}_\theta \, \mathbb{E}_{x_T \sim \tilde{\pi}_f(\cdot \mid \theta)}\big[R^0(x_T, \theta)\big]. \qquad (*)$$

The case $T = 1$ (zero-shot) follows verbatim by replacing $x_T$ with the single-step $x$. $\qquad \square$

(a) *Heat 1D*

(b) *Euler 1D*

(c) *NS Transient 2D*

Figure 8: Test results of our best neural network for each task. The plots from left to right respectively mean: (left) soft utility signal of true RMSE loss *vs.* soft utility signal of predicted RMSE loss, (middle) true cost *vs.* predicted cost, (right) distribution in the cost-utility space of predicted *vs.* true points.

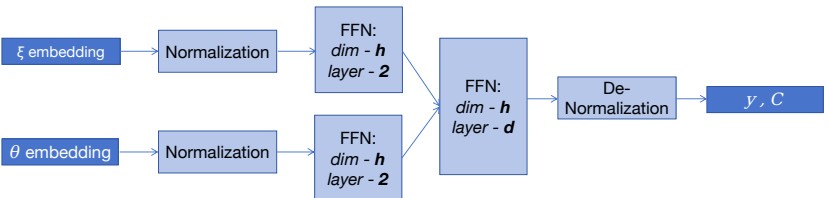

Figure 9: Neural-Network Structure

In this work, we define the soft utility function $f(r)$ as follows:

$$f(r) = \begin{cases} 1.0 & \text{if } d \leq \epsilon \\ \alpha e^{-\beta(r-1)^\gamma} + (1-\alpha)\left(\frac{1}{1+\omega(r-1)^\delta}\right) & \text{if } d > \epsilon, \end{cases} \tag{10}$$

where $r = \frac{d}{\epsilon}$. The parameters are set to $\alpha = 0.6$, $\beta = 0.43$, $\gamma = 1.5$, $\omega = 0.3$, and $\delta = 2.2$.

This function is designed so that the utility value drops to approximately 0.5 when the distance $d$ is double the tolerance $\epsilon$ (i.e., $r = 2$), and it decays rapidly towards zero as the distance increases further, becoming negligible for distances approaching $10\epsilon$ (i.e., $r = 10$). A plot of $f(r)$ is shown in Figure 10.

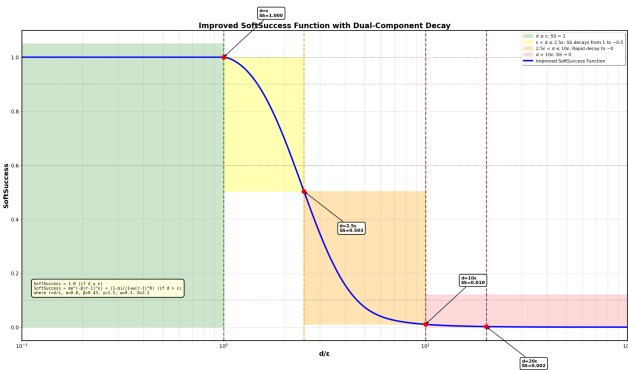

Figure 10: Plot of the soft utility function $f(r)$. The function maintains a maximum utility of 1.0 for normalized distances $r \leq 1$, drops to approximately 0.5 at $r = 2$, and rapidly decays towards zero for larger values of $r$.

## E  PROMPTS USED IN AGENT FRAMEWORK

---

**Prompt Example for *Euler 1D* Single-Turn w/. Scenario Setting**

**Instruction**

Your task is to find the optimal parameter, solving the 1D Euler equations for compressible inviscid flow, using a 2nd order MUSCL scheme with Roe flux and generalized superbee limiter. This serves as a simplified model for compressible fluid dynamics. You should try to minimize the total cost incurred by function calls, but your primary goal is to successfully meet the convergence criteria. You should always use the tool call function to finish the problem.

Workflow: n_space (Number of grid cells) determines the spatial discretization resolution: $\Delta x = L/n\_space$ where L is the domain length. You may **only** change 'n_space'. The value of k is **-1.0**, beta is **1.0**, cfl is **0.25**. **You must not change them!** You have only one opportunity to choose an optimal value for n_space. No trial-and-error or iterative optimization is permitted. Your goal is to select a value that provides adequate spatial resolution while keeping computational cost reasonable.

Step 1: Make your best **one-shot** guess for n_space.
Step 2: Call the Convergence Test Function and check if converged.
Step 3: Output final answer with no further tool calls.

**Input**

QID: 1
Problem: Euler 1D Equations with 2nd Order MUSCL-Roe Method
This simulation solves the 1D Euler equations for compressible inviscid flow, using a 2nd order MUSCL scheme with Roe flux and generalized superbee limiter:
Conservative form:

$$\frac{\partial \mathbf{U}}{\partial t} + \frac{\partial \mathbf{F}(\mathbf{U})}{\partial x} = 0$$

---

Where the conservative variables and flux are:

$$\mathbf{U} = \begin{pmatrix} \rho \\ \rho u \\ \rho E \end{pmatrix}, \quad \mathbf{F} = \begin{pmatrix} \rho u \\ \rho u^2 + p \\ u(\rho E + p) \end{pmatrix}$$

Primitive variables:

- $\rho$ = density
- $u$ = velocity
- $p$ = pressure
- $E$ = specific total energy

Equation of state:

$$p = (\gamma - 1)\rho \left( E - \frac{u^2}{2} \right)$$

where $\gamma$ is the ratio of specific heats.

Spatial Discretization: The spatial discretization uses MUSCL reconstruction with blending parameter $k$:

$$\mathbf{U}^L_{j+\frac{1}{2}} = \mathbf{U}_j + \frac{1+k}{4}\psi(r_j)(\mathbf{U}_{j+1} - \mathbf{U}_j)$$

$$\mathbf{U}^R_{j+\frac{1}{2}} = \mathbf{U}_{j+1} - \frac{1+k}{4}\psi(r_{j+1})(\mathbf{U}_{j+2} - \mathbf{U}_{j+1})$$

where $k$ is a blending coefficient between central ($k = 1$) and upwind ($k = -1$) scheme, and $\psi(r)$ is the slope limiter function.

Slope Limiting: The slope limiter uses a generalized superbee limiter:

$$\psi(r) = \max\left[0, \max\left[\min(\beta r, 1), \min(r, \beta)\right]\right]$$

where $\beta$ is the limiter parameter controlling dissipation.

The slope ratio $r$ at interface $j$ is defined as:

$$r_j = \frac{\mathbf{U}_{j+1} - \mathbf{U}_j}{\mathbf{U}_{j+2} - \mathbf{U}_{j+1}}$$

This ratio indicates the local non-smoothness, which will be the input into the slope limiter to achieve the TVD condition.

Flux Computation: The interface flux is computed using the Roe approximate Riemann solver:

$$\mathbf{F}_{j+\frac{1}{2}} = \frac{1}{2}\left[\mathbf{F}(\mathbf{U}^L) + \mathbf{F}(\mathbf{U}^R)\right] - \frac{1}{2}|\mathbf{A}|(\mathbf{U}^R - \mathbf{U}^L)$$

where $|\mathbf{A}|$ is the Roe matrix with Roe-averaged quantities.

Initial condition cases:

- sod: Left: $\rho = 1.0, u = 0.0, p = 1.0$; Right: $\rho = 0.125, u = 0.0, p = 0.1$
- lax: Left: $\rho = 0.445, u = 0.6977, p = 3.528$; Right: $\rho = 0.5, u = 0.0, p = 0.571$
- mach_3: Left: $\rho = 3.857, u = 0.92, p = 10.333$; Right: $\rho = 1.0, u = 3.55, p = 1.0$

Parameter Information:

- cfl: Courant-Friedrichs-Lewy number, $CFL = \frac{(|u|+c)\Delta t}{\Delta x}$ where $c = \sqrt{\gamma p/\rho}$ is the speed of sound
- beta: Limiter parameter for generalized superbee
- k: Blending parameter between central and upwind fluxes
- n_space: Number of grid cells for spatial discretization, determines spatial resolution: $\Delta x = L/n\_space$

Physical Parameters:

- Domain length: 1.0

- Gamma (ratio of specific heats): 1.4
- Case: sod

Convergence Check:

- Errors between the simulation based on your solution and the simulation based on the self-refined solution are computed to assess convergence.
- Convergence is confirmed if the following validation criteria are satisfied.

Validation Criteria:

- **Current Problem Precision Level**: HIGH
- **Required RMSE Tolerance**: $\leq 0.01$
- Relative RMSE must meet this tolerance compared to self-refined solution
- Positivity preservation: pressure and density must remain positive at all times
- Shock speed consistency: pressure gradients should not exceed physical bounds

**Available functions:**

Function Name: euler_1d_check_converge_n_space

Description: Conduct a 1D Euler PDE simulation and evaluate its spatial convergence by doubling n_space. It returns the following results:

- RMSE: float
- is_converged: boolean
- accumulated_cost: integer
- The cost of the solver simulating the environment: integer
- The cost of the solver verifying convergence (This will not be included in your accumulated_cost): integer
- metrics1: object
- metrics2: object

Parameters:

- cfl (float): CFL number
- beta (float): Limiter parameter for generalized superbee
- k (float): Blending parameter for MUSCL reconstruction
- n_space (integer): Current number of grid cells for spatial discretization

Required parameters: cfl, beta, k, n_space **Design-Value History**

```
Below are some previous n_space values and their simulation accuracy
↪   and efficiency indicators. The values are arranged in ascending
↪   order based on accuracy, where higher values indicate a closer
↪   simulation result to ground truth. The efficiency indicator is
↪   also important, where higher values mean a more cost-efficient
↪   n_space choice.

<n_space> 240 </n_space>
 Accuracy Indicator:
0.9834
 Efficiency Indicator:
1.1479

<n_space> 512 </n_space>
 Accuracy Indicator:
1.0000
 Efficiency Indicator:
0.2717
```

```
<n_space> 400 </n_space>
 Accuracy Indicator:
1.0000
 Efficiency Indicator:
0.4255

<n_space> 300 </n_space>
 Accuracy Indicator:
1.0000
 Efficiency Indicator:
0.7290

<n_space> 288 </n_space>
 Accuracy Indicator:
1.0000
 Efficiency Indicator:
0.7897

<n_space> 260 </n_space>
 Accuracy Indicator:
1.0000
 Efficiency Indicator:
0.9707

<n_space> 258 </n_space>
 Accuracy Indicator:
1.0000
 Efficiency Indicator:
0.9865

<n_space> 257 </n_space>
 Accuracy Indicator:
1.0000
 Efficiency Indicator:
0.9946

<n_space> 256 </n_space>
 Accuracy Indicator:
1.0000
 Efficiency Indicator:
1.0027

<n_space> 252 </n_space>
 Accuracy Indicator:
1.0000
 Efficiency Indicator:
1.0364

Output final answer in the requested format with a new n_space value
↪  that is different from all values above. You should first ensure
↪  an accurate simulation by achieving 1.0 in accuracy indicator,
↪  then gradually increase efficiency by choosing a coarser n_space
↪  value.
```

**Prompt Example for *Euler 1D* Single-Turn w/o Scenario Setting**

**Instruction**

Your task is to optimize a one-dimensional black-box function with a given parameter. You will be prompted with a list of history of parameter and values, where values include an accuracy indicator and success indicator. You are required to first optimize accuracy until it reaches

1.0, then optimize efficiency for as high as possible. The parameter in history will start with <n_space>and end with </n_space>. Please return a parameter value different from all values given in the history that you think will optimize the function value as requested. Please return your answer by starting with <n_space>and ending with </task>as well. **You may NOT use any form of prior knowledge, and treat all parameter names, function names, etc. as purely arbitrary.**

**Input**

**Design-Value History**

```
Below are some previous n_space values and their simulation accuracy
↪   and efficiency indicators. The values are arranged in ascending
↪   order based on accuracy, where higher values indicate a closer
↪   simulation result to ground truth. The efficiency indicator is
↪   also important, where higher values mean a more cost-efficient
↪   n_space choice.

<n_space> 240 </n_space>
 Accuracy Indicator:
0.9834
 Efficiency Indicator:
1.1479

<n_space> 512 </n_space>
 Accuracy Indicator:
1.0000
 Efficiency Indicator:
0.2717

<n_space> 400 </n_space>
 Accuracy Indicator:
1.0000
 Efficiency Indicator:
0.4255

<n_space> 300 </n_space>
 Accuracy Indicator:
1.0000
 Efficiency Indicator:
0.7290

<n_space> 288 </n_space>
 Accuracy Indicator:
1.0000
 Efficiency Indicator:
0.7897

<n_space> 260 </n_space>
 Accuracy Indicator:
1.0000
 Efficiency Indicator:
0.9707

<n_space> 258 </n_space>
 Accuracy Indicator:
1.0000
 Efficiency Indicator:
0.9865

<n_space> 257 </n_space>
  Accuracy Indicator:
```

```
1.0000
 Efficiency Indicator:
0.9946

<n_space> 256 </n_space>
 Accuracy Indicator:
1.0000
 Efficiency Indicator:
1.0027

<n_space> 252 </n_space>
 Accuracy Indicator:
1.0000
 Efficiency Indicator:
1.0364

Output final answer in the requested format with a new n_space value
↪  that is different from all values above. You should first ensure
↪  an accurate simulation by achieving 1.0 in accuracy indicator,
↪  then gradually increase efficiency by choosing a coarser n_space
↪  value.
```

Statistics Appended to Prompts

Below is the statistical information extracted from historical samples concerning task values and scores concerning accuracy and efficiency:
1. **Overall scale of {task}**
- Typical x range: {x_range}
- Mean x value: {x_mean:.4f}
2. **Best and worst observed samples**
- Best sample: x = {x_best}, y = {y_best:.4f}
- Worst sample: x = {x_worst}, y = {y_worst:.4f}
3. **Global trend between x and y**
- Pearson correlation between x and y: {pearson_corr:.4f}
- Fitted model for global trend: {fitted_model_description}
Please respond strictly according to the json format specified before.
Return your answer in JSON format.

## F  DETAILED RESULTS

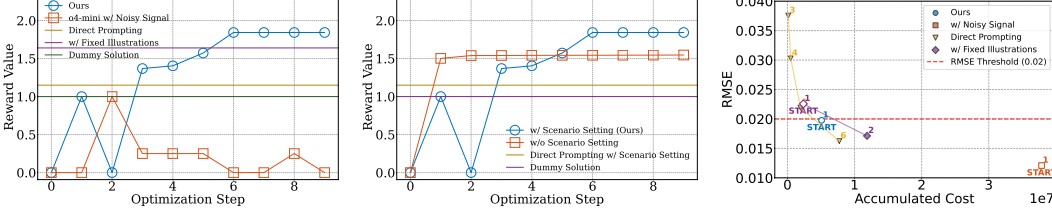

(a) An exemplar optimization trajectory in Single-Turn setting for *CASCO-Agent* with vs. without scenario setting.

(b) An exemplar optimization trajectory in Single-Turn setting for *CASCO-Agent* with various signals.

(c) An exemplar optimization trajectory in Mingle-Turn setting for *CASCO-Agent* with various signals.

Figure 11: **Case studies** for ablations, with base model OpenAI o4-mini.

Full results of the comprehensive benchmark are presented in table 3. The case studies as introduced in 4.5 are shown in Figure 11.

The full ablation results are presented in table 4. Our ablations on surrogate neural network and prior knowledge are conducted on *Euler 1D* 's medium precision level tasks; we report reward $R^0, R^m$ and success rates $P^0, P^m$ for both Single-Turn and Multi-Turn settings. Our base model is fixed as

Table 3: Main evaluation results in both Single-Turn and Multi-Turn settings. Values in each box is the mean of tasks evaluated in three precision levels. Note that we report both reward $R^0$, $R^m$ and for-reference quantities $P^0$ and $P^m$. Values in bold font are the best-achieving ones, and values with ↑ indicate a significant rise compared to direct prompting.

(a) Single-Turn *CASCO-Agent*

| Method | Base Model | Heat 1D | | Euler 1D | | NS Transient 2D | |
|---|---|---|---|---|---|---|---|
| | | $R^0$ | $P^0$ | $R^0$ | $P^0$ | $R^0$ | $P^0$ |
| BO (Nogueira, 2014) | – | 0.253 | **1.000** | 0.464 | 0.125 | 0.814 | 0718 |
| Base LLM | Llama3.2-3B-Instruct | 0.288 | 0.347 | 0.698 | 0.174 | 0.052 | 0.151 |
| | Qwen-8B | 0.412 | 0.633 | 0.642 | 0.268 | 0.342 | **1.000** |
| | o4-mini | 0.362 | 0.253 | 0.501 | 0.301 | 0.565 | 0.516 |
| *CASCO-Agent* (Ours) | Llama3.2-3B-Instruct | 0.950↑ | 0.773 ↑ | 0.939 ↑ | 0.516 ↑ | 1.591↑ | 0.785↑ |
| | Qwen-8B | 0.853↑ | 0.759 ↑ | 0.897 ↑ | **0.789** ↑ | **1.813**↑ | 0.702 |
| | o4-mini | **1.239** ↑ | 0.679 ↑ | **1.764** ↑ | 0.733 ↑ | 0.842 ↑ | 0.536 |

(b) Multi-Turn *CASCO-Agent*

| Method | Base Model | Heat 1D | | Euler 1D | | NS Transient 2D | |
|---|---|---|---|---|---|---|---|
| | | $R^m$ | $P^m$ | $R^m$ | $P^m$ | $R^m$ | $P^m$ |
| BO (Nogueira, 2014) | – | 0.290 | **1.000** | 0.496 | 0.625 | 0.517 | 0.766 |
| Base LLM | Llama3.2-3B-Instruct | 1.060 | 0.837 | 0.328 | 0.531 | 1.232 | 0.448 |
| | Qwen-8B | 1.613 | 0.756 | 1.511 | 0.421 | 0.662 | 0.861 |
| | o4-mini | 1.960 | 0.960 | 1.135 | 0.392 | 0.991 | 0.674 |
| OPRO (Yang et al., 2023) | Llama3.2-3B-Instruct | 0.170 | 0.917 | 0.290 | 0.600 | 0.275 | 0.877 |
| | Qwen-8B | 0.217 | 0.917 | 0.323 | **0.680** | 0.326 | **1.000** |
| | o4-mini | 0.241 | 0.917 | 0.974 | 0.520 | 0.957 | **1.000** |
| *CASCO-Agent* (Ours) | Llama3.2-3B-Instruct | 1.204 ↑ | 0.946 ↑ | 1.339 ↑ | 0.572 | 1.435 | 0.925 ↑ |
| | Qwen-8B | 1.760 | 0.900 ↑ | 1.359 | 0.624 ↑ | 1.535 ↑ | 0.944 |
| | o4-mini | **1.981** | 0.986 | **1.571** ↑ | 0.443 | **1.538** ↑ | 0.972 ↑ |

o4-mini. Note that although *CASCO-Agent* without Physics prior is achieving a higher mean reward in Single-Turn setting, its success rate is much lower than our method, indicating its frequent choice of coarse designs that leads to high reward in only a few tasks. We argue that this is a form of reward hacking as it contradicts with our expectation to carry out experiments correctly and efficiently.

Table 4: Ablation results averaged over all tasks.

| Setting | Single-Turn Setting | | Multi-Turn Setting | |
|---|---|---|---|---|
| | $R^0$ | $P^0$ | $R^m$ | $P^m$ |
| *CASCO-Agent* (Ours) | 0.571 | **0.708** | **0.834** | 1 |
| *CASCO-Agent* w/ Sparse Surrogate Signal | 0.42 | 0.5 | 0.635 | 0.875 |
| *CASCO-Agent* w/ Random Signal | 0.142 | 0.583 | 0.426 | 0.583 |
| *CASCO-Agent* w/ In-Context Signal | 0.42 | 0.5 | 0.572 | **0.958** |
| *CASCO-Agent* w/o Physics Prior | **0.595** | 0.152 | 0.475 | 0.375 |
| Direct Prompting | 0.096 | 0.125 | 0.116 | 0.167 |

# G  IMPLEMENTATION DETAILS

We include a detailed and explicit description of all baseline implementation in this section.

**Bayesian Optimization**

**Surrogate Model.** We use Gaussian Process regressor with Matern Kernel. We initialize the kernel with smoothness parameter = 2, length scale = 1.

**Optimizer for Surrogate Model Training.** We use L-BFGS (Liu and Nocedal, 1989) implemented in sk-learn (Pedregosa et al., 2018). We use

1. $\alpha = 0.6$, i.e. the amount of observation noise added to the diagonal of the covariance matrix during training
2. $n\_restarts\_optimizer = 10$, i.e. number of times to restart L-BFGS.
3. $length\_scale\_bounds = (1e^{-2}, 1e^{1})$, i.e. the admissible range for the ARD length-scale parameters during hyperparameter optimization.

**Acquisition Function.** We use UCB with confidence width (kappa) = 2.576

**Direct Prompting**

temperature = 0.8, maximum new tokens = 64, top-K = 50, top-P = 0.9

**OPRO**

temperature = 0.8, maximum new tokens = 64, top-K = 50, top-P = 0.9, number of iterations = 5, number of generation branches each step = 4

*CASCO-Agent*

temperature = 0.8, maximum new tokens = 64, top-K = 50, top-P = 0.9, number of iterations = 5, number of generation branches each step = 4

## H  EXPERIMENTAL BUDGETS

We list computational budgets in Table 5 and Table 6

Table 5: Computational budgets of methods in single-turn setting. [a]: Using the best proposal selected from 20 sampled responses (temperature = 0.7). [b]: Sampling 4 responses (temperature = 0.7) for 5 optimization steps. [c]: Computation time excluding simulator runtime, averaged across all problems.

| Method | Runtime per problem [c] | # of Calls to simulators per problem | # of calls to LLMs per problem |
|---|---|---|---|
| BO | 16.7s | 1 | N/A |
| Direct Prompting | 122.6s | 1 | 20 [a] |
| Ours | 140.52s | 1 | $5 \times 4 = 20$ [b] |

Table 6: Computational budgets of methods in multi–turn setting. [a]: Termination of the multi-turn query process is determined by the agent; the maximum number of allowed steps is 10. [b]: t denotes the number of steps taken before the agent decides to terminate for each trajectory; 4 trajectories are sampled for all methods (temperature = 0.7). [c]: Computation time excluding simulator runtime, averaged across all problems. [d]: Lower values arise because agents within the CAED-Agent framework tend to converge earlier and thus terminate the multi-turn process in fewer steps.

| Method | Runtime per problem [c] | # of Calls to simulators per problem | # of calls to LLMs per problem |
|---|---|---|---|
| BO | 34.71s | 10 | N/A |
| Direct Prompting | 96.80s | $1 \sim 10$[a] | $4t$[b] |
| OPro | 115.58s | $1 \sim 10$[a] | $4t$ [b] |
| Ours | 60.57s [d] | $1 \sim 10$[a] | $4t$[b] |

## I  ADDITIONAL ABLATION EXPERIMENTS

**Using a separate surrogate model for evaluation is more effective than providing samples directly to the LLM.** To illustrate necessity of introducing a separately trained neural surrogate to

evaluate candidate designs, instead of simply exposing the LLM to the training examples (or summary statistics) through prompting or fine-tuning, we designed an additional set of experiments. Specifically, we compare the original CAED-Agent against:

1. Few-shot prompting with 5/10/20 in-context illustrations (Fewshot-5/10/20). As in our original ablation, the illustrations are randomly sampled from samples with the same conditions in the training dataset, arranged in ascending order, and appended to the prompt we previously used. (See Appendix E.)

2. Direct Prompting with statistics derived from all training samples for the neural surrogate. (DP+stats) The following statistics are provided: variable range, best/worst samples, Pearson correlation, fitted model (using quadratic regression) descriptions. See the appended statistics under "Statistics Appended to Prompts" in Appendix E.

3. CAED-Agent appended with 5/10/20 in-context illustrations (CAED-Fewshot-5/10/20). The samples are chosen in the same manner as (1), and appended to the prompt. The querying of neural surrogate and iterative update of the illustrations are the same as our original method.

Table 7: Ablation studies on few-shot prompting and direct prompting with statistics. Done on euler_1d n_space, single-turn.

|                 | CAED-Agent | Fewshot | DP+Stats | CAED-Fewshot |
|-----------------|------------|---------|----------|--------------|
| 5 Illustrations |            | 0.476   |          | **0.622**    |
| 10 Illustrations| 0.571      | 0.42    | 0.289    | **0.635**    |
| 20 Illustrations|            | 0.365   |          | **1.058**    |

See Table 7 for results: (1) Providing only summary statistics consistently underperforms. Simulation-design tuning is a fine-grained task requiring relational information beyond what statistical descriptors can convey. (2) Few-shot prompting variants also fail to surpass our method, and the variant using an extended context (20 illustrations) performs even worse. Our failure-mode analysis suggests that LLMs tend to copy solutions from illustrations, leading to suboptimal proposals. (3) Augmenting CAED-Agents prompts with few-shot illustrations improves the agents exploratory behavior, yielding the strongest performance among all tested configurations.

