# OpenReview forum: "CAED-Agent: an Agentic Framework to Automate Simulation-Based Experimental Design"
_ICLR.cc/2026/Conference — Submitted to ICLR 2026_

### Official Review · Reviewer_toXH · 2025-10-25

**Soundness:** 2
**Presentation:** 3
**Contribution:** 2
**Rating:** 4
**Confidence:** 4

**Summary:**

This paper introduces CAED-Agent, an agentic framework that aims to improve the cost-efficiency of simulation-based experimental design. The key idea is to pair a large language model’s inference-time reasoning with a lightweight neural network that predicts cost and utility signals, so that the LLM can make informed, cost-aware decisions without excessive simulator calls. The authors test the approach on three physics simulation environments (1D heat conduction, 1D Euler, and 2D Navier–Stokes) and show that CAED-Agent achieves better success rates and efficiency than Bayesian optimization and other LLM-based optimizers.

**Strengths:**

- The paper tackles a well-motivated and practically important problem: how to make LLM-driven scientific agents more cost-efficient and aware of computational budgets. It clearly identifies two major pain points of current approaches: lack of cost awareness and inference-scaling inefficiency.
- The problem formulation is clean, and the proposed surrogate-signal design is simple yet effective. The experimental setup, while small-scale, convincingly shows that cost-aware signals can meaningfully improve sample efficiency.
- Overall, the work provides a reasonable step toward more practical LLM-based experiment design agents, and the results are consistent with the claimed motivation.

**Weaknesses:**

- The assumption that “small fully connected neural networks can learn the cost and utility functions well” feels overly strong and only holds because the experiments are limited to very low-dimensional (1D–2D) settings. It’s unclear whether the same idea would scale to higher-dimensional or more realistic simulation problems.
- The design choice of adding a small neural network to guide the LLM is not fully justified. If the cost and utility mappings are simple enough for a tiny NN to learn, one might question why the LLM itself cannot capture such patterns through proper prompting or fine-tuning. The rationale for separating the learning responsibilities between the LLM and the NN needs clearer theoretical or empirical backing.
- The related-work discussion could be deeper. There is rich literature in BO that can leverage prior knowledge from related domains. A simple Google search will return many papers. E.g., Pre-trained Gaussian processes for Bayesian optimization (Wang et al. 2024).
- The clarity of presentation could also improve: a few typos (e.g., “benifit” in line 091), small fonts and dense descriptions in Figure 2 make it hard to follow.
- Finally, the evaluation is confined to toy problems; no evidence is provided that CAED-Agent can handle realistic scientific simulations where costs and outcomes are high-dimensional or noisy.

**Questions:**

- It would help to clarify the exact claim that “Bayesian optimization cannot generalize across problem variations.” BO’s limitations usually stem from surrogate transfer, not from the BO framework itself.
- The cost function in Eq. (2) seems to depend on y (simulation outputs), but intuitively cost should depend only on x and θ; please clarify this dependence.
- Future work could explore whether the cost-aware feedback can be incorporated directly into the LLM through reinforcement-style prompting or few-shot demonstrations, possibly removing the need for a separate NN surrogate.

---

> ### Author Response · Authors · 2025-11-22
> **Response to Reviewer toXH (Part 1)**
>
> We thank the reviewer for the constructive suggestions and acknowledgement of our work's value.
>
> > Q1 - It would help to clarify the exact claim that “Bayesian optimization cannot generalize across problem variations.” ...
>
> We clarify BO’s limitations with the following points:
> - **Bayesian optimization operates on a fixed, vectorized numeric search space**, making it rigid when problem variations involve structural or semantic changes. Multi-objective tasks require scalarization into a single numeric objective (Knowles, 2006; Marler, 2004), and tasks described in natural language (e.g., “single vs. tandem airfoil wing”) may be difficult to parameterize; these scenarios will require case-specific, costly manual transferring efforts. In contrast, LLMs can directly ingest and generalize over such textual or symbolic problem descriptions without re-engineering the parameter space.
> - **Experts can use non-numerical or verbal heuristics** (e.g., “decrease the time-step if you see jittering in simulations”). Such heuristics are difficult to encode into BO kernels or acquisition functions (Shahriari et al., 2016), whereas LLMs can naturally interpret and apply these linguistic rules, enabling cross-task transfer that BO cannot obtain without manual redesign.
> - **Although pre-trained Gaussian Process surrogates exist, they still lack transferable prior knowledge** and therefore show poor early-phase sample efficiency (Snoek et al., 2012; Wistuba et al., 2018; Wang et al., 2024). LLMs bring rich domain priors at zero-shot, improving initial exploration when simulation setups vary.
>
> [1] Knowles, J. (2006). ParEGO: A hybrid algorithm with on-line landscape approximation for multiobjective optimization. In IEEE Transactions on Evolutionary Computation, 10(1), 50–66.
>
> [2] Marler, R. T., & Arora, J. S. (2004). Survey of multi-objective optimization methods for engineering. In Structural and Multidisciplinary Optimization, 26, 369–395.
>
> [3] Shahriari, B., Swersky, K., Wang, Z., Adams, R. P., & de Freitas, N. (2016).Taking the Human Out of the Loop: A Review of Bayesian Optimization. In Proceedings of the IEEE, 104(1), 148–175.
>
> > W1 - The assumption that “small fully connected neural networks can learn the cost and utility functions well” feels overly strong ....
>
> > W5 - Finally, the evaluation is confined to toy problems....
>
> We would like to emphasize that our study represents one of the first attempts at cost-aware simulation-based experimental design. Even within these simplified settings, our experiments demonstrate that naïve strategies for using LLMs fail (Figure-3; Table-2, Columns 5–6 of our paper).
>
> Recent work on LLM-automated experiment design (Chan et al., 2025; Chen et al., 2025) implicitly assumes that all forward simulations proposed by LLMs/agents are both reliable and efficient. Our results—even on comparatively simple scenarios—show that this assumption does not hold, especially when the system encounters novel or previously unseen simulation configurations. Thus, we believe our practical contribution lies in revealing the often-overlooked interplay between simulation cost and simulation accuracy in LLM-driven design, and in providing a principled approach for addressing this challenge.
>
> That said, we fully acknowledge the value of more realistic tests. In the revised version, we will include experiments on additional high-complexity systems, such as the Hasegawa–Mima Nonlinear Equation solved with a pseudo-spectral method. This system exhibits strong nonlinear mode coupling and incurs substantial computational cost. Results are currently being produced.
>
> In addition, to correspond to *realistic simulation problems* where the process is often chaotic, we conducted an ablation study on a modified Euler 1D simulator with added Gaussian noise (1% of the standard derivation of the training dataset). Results show the performance drop for our method is less significant than baselines, proving our method has some robustness to realistic scenarios. The baselines suffer a significant performance drop: around 80% for DP, 69% for BO.
>
> | Method / Task                 | Direct Prompting | Bayesian Optimization | CAED-Agent w/ o4-mini |
> |------------------------------|------------------|------------------------|------------------------|
> | Euler_1d n_space w/. noise   | 0.034            | 0.122                  | 0.497                  |
> | Euler_1d n_space             | 0.171            | 0.391                  | 0.571                  |

---

> ### Author Response · Authors · 2025-11-22
> **Response to Reviewer toXH (Part 2)**
>
> > W2 - The design choice of adding a small neural network to guide the LLM is not fully justified. ... needs clearer theoretical or empirical backing.
>
> We thank the reviewer for this comment. We provide the following experiments, which is also updated to Section 4.5 of the pdf:
>
> **Using a separate surrogate model for evaluation is more effective than providing samples directly to the LLM.**
>
> To illustrate necessity of introducing a separately trained neural surrogate to evaluate candidate designs, instead of simply exposing the LLM to the training examples (or summary statistics) through prompting or fine-tuning,  we designed an additional set of experiments. Specifically, we compare the original CAED-Agent against:
>
> - Few-shot prompting with 5/10/20 in-context illustrations (Fewshot-5/10/20). As in our original ablation, the illustrations are randomly sampled from samples with the same conditions in the training dataset, arranged in ascending order, and appended to the prompt in Appendix E.
>
> - Direct Prompting with statistics derived from all training samples for the neural surrogate. (DP+stats) We provide the following statistics: variable range, best/worst samples, Pearson correlation, fitted model (using quadratic regression) descriptions.
>
> - CAED-Agent appended with 5/10/20 in-context illustrations (CAED-Fewshot-5/10/20). The samples are chosen in the same manner as (1), and appended to the prompt. The querying of neural surrogate and iterative update of the illustrations are the same as our original method.
>
> | Illustrations     | CAED-Agent | Fewshot | DP+Stats | **CAED-Fewshot** |
> |-------------------|------------|---------|----------|------------------|
> | 5 Illustrations   | 0.571      | 0.476   | 0.289    | **0.622**        |
> | 10 Illustrations  |            | 0.42    |          | **0.635**        |
> | 20 Illustrations  |            | 0.365   |          | **1.058**        |
>
> **Table-1: Experiments done on euler_1d n_space, single-turn.**
>
> Results show that (1) Few-shot prompting variants fail to surpass our method, and the variant using an extended context (20 illustrations) performs even worse. Our failure-mode analysis suggests that LLMs tend to copy solutions from illustrations, leading to suboptimal proposals. (2) Augmenting CAED-Agent’s prompts with few-shot illustrations improves the agent’s exploratory behavior, yielding the strongest performance among all tested configurations.
>
> > W3-The related-work discussion could be deeper. .... E.g., Pre-trained Gaussian processes for Bayesian optimization (Wang et al. 2024).
>
> Thank you, we’ll provide a more holistic review of domain-knowledge BO in a revised manuscript, adding works including:
>
> *Martin Wistuba, Josif Grabocka. “Few-Shot Bayesian Optimization with Deep Kernel Surrogates.” ICLR 2021.*
>
> *Zi Wang, George E. Dahl, Kevin Swersky, Chansoo Lee, Zachary Nado, Justin Gilmer, Jasper Snoek, Zoubin Ghahramani. “Pre-trained Gaussian Processes for Bayesian Optimization.” JMLR 25(212):1-83, 2024*
>
> *A. Souza et al. “Prior-guided Bayesian Optimization.” Proceedings of the 4th Workshop on Meta-Learning at NeurIPS 2020.*
>
> *Zhou Fan, Xinran Han, Zi Wang. “HyperBO+: Pre-training a universal prior for Bayesian optimization with hierarchical Gaussian processes.” arXiv 2022.*
>
> But we maintain our claim of novelty after reviewing the above works. While prior-aware BO improves sample efficiency, they still do not target cost-aware experimental design nor solve the shortcomings of BO we list in part 1 of our response (to Q1).
>
> > W4-The clarity of presentation could also improve: a few typos (e.g., “benifit” in line 091), small fonts and dense descriptions in Figure 2 make it hard to follow.
>
> Thank you for mentioning this oversight, we will double-check the captions, and increase clarity of the figure in a revised manuscript.
>
> > Q2-The cost function ... clarify this dependence.
>
> In Eq. (1), y is defined as
> \begin{equation}
> y = \mathcal{F}(x, \theta), \qquad x \in \mathcal{X},\; \theta \in \Theta
> \end{equation},
> i.e, y is also dependant on $x$ and $\theta$, given a forward process $\mathcal{F}$. We chose to use y as the input for reward function because the success metric solely depends on y. We will revise our formula in the revised version.
>
> > Q3-Future work could explore whether the cost-aware feedback can be incorporated directly into the LLM ....
>
> We agree with this direction of future work. However, we note that (1) for post-training, using neural network (instead of real simulators) is the most efficient method to either provide reward signals (e.g, for RL) or accumulate good examples (e.g., for SFT) (2) according to experiments we provided earlier, simply adding few-shot illustrations fails to surpass the performance with neural surrogates. More sophisticated future work is needed for this direction.

---

### Official Review · Reviewer_WNEs · 2025-10-26

**Soundness:** 3
**Presentation:** 3
**Contribution:** 2
**Rating:** 4
**Confidence:** 3

**Summary:**

This paper presents a method for automating parameter tuning in computational simulations that must balance accuracy and computational cost. It introduces the concept of cost-aware experimental design (CAED) and proposes CAED-Agent, which combines a large language model with a lightweight neural surrogate trained to predict simulation utility and cost. The surrogate provides feedback signals that the LLM uses in context to iteratively generate improved design parameters without retraining.

The framework is evaluated on three physics-based simulations, including Heat 1D, Euler 1D, and Navier–Stokes 2D, and compared with Bayesian Optimization and Optimization by Prompting (OPRO). The results show that CAED-Agent generally achieves better cost–utility trade-offs and more stable optimization behavior than these baselines, particularly in multi-turn settings, while requiring fewer full simulator evaluations.

**Strengths:**

1. The formulation of the cost-aware experimental design problem is novel and meaningful, addressing a problem that has been underexplored.
2. The use of a smaller surrogate model to provide continuous cost–utility feedback to a large language model is a well-motivated idea, offering an efficient way to utilize prior knowledge or existing data when available.
3. The writing is generally clear and easy to follow.

**Weaknesses:**

1. The accuracy of the surrogate model is important, as shown in the ablation study. This method requires specific data for each experimental task, which is somewhat unrealistic. In the experiments, more than 4,000 samples were used to train such a model (in 2 out of the 3 tasks), making it unsuitable for few-shot scenarios where experiments are expensive.

2. The statements “outperforms both ... by significant margins” and “Through experiments on three physics simulator environments, each with varying environmental settings and precision requirements, we demonstrated that CAED-Agent consistently outperforms both classical Bayesian optimization baselines and state-of-the-art LLM-based optimizers” are vague. The paper should clarify the metrics on which these claims are based. For example, in terms of pass rate, CAED-Agent does not consistently outperform baseline methods.

3. Some experimental details are missing, such as the hyperparameters of the LLMs and the settings of baseline methods.

4. The experiments are somewhat narrow in scope. All evaluations are conducted on relatively low-dimensional 1D and 2D PDE simulations with a small number of design variables (e.g., grid size, CFL number).

**Questions:**

1. Please refer to the weaknesses.
2. How beneficial is it to use a neural network to model the prior knowledge? Would incorporating simple statistics derived from the existing samples (e.g., the possible ranges of the design variables) achieve a similar effect?

---

> ### Author Response · Authors · 2025-11-22
> **Response to Reviewer WNEs**
>
> We thank the reviewer for acknowledging our strengths and providing valuable comments.
>
> > W1-The accuracy of the surrogate model is important, ... specific data for each experimental task, which is somewhat unrealistic.  ... unsuitable for few-shot scenarios where experiments are expensive.
>
> Regarding size of the training dataset, we provide the following justifications of our method in practice:
> - The offline training set usually exists in the lab seeking to use LLM for automation. For example, simulator logs with parameter choices, corresponding outcomes, and wall time are usually recorded in simulation runs. Using these the training targets can be computed directly.
> - The cost for most simulators is only dependent on a few key factors, e.g. mesh cell numbers, time step sizes, integration order etc. This limits the input dimension variety and hence the sampling space.
> - In addition, we incorporated the rough complexity analysis as prior and only trained the NN to learn the residual between that and the real incurred cost, which can further reduce the training set size. As shown in our ablation study below on different training set sizes, this design allows our method to outperform Bayesian Optimization (with full 4K dataset) using as few as 500 samples. (See below table).
>
> | Method               | 100 samples | 500 samples | 2000 samples | full samples (~4k) |
> |----------------------|-------------|-------------|--------------|---------------------|
> | **CAED-Agent (Ours)** | 0.239       | 0.471       | 0.521        | **0.571**           |
> | BO                   | 0.418       | 0.392       | 0.414        | 0.391               |
> | Direct Prompting     | 0.171       |             |              |                     |
>
> **Table-1: Experiments done on euler_1d n_space, single-turn.**
>
> > W2&3 - The statements “outperforms ...” are vague. ...  Some experimental details are missing, ...
>
> Thanks for your suggestions; please refer to Appendix G of our updated pdf for a detailed explanation of our hyper parameters and experimental details. For metrics, please refer to Section 4.3 of our paper.
>
> We stress that our optimization goal is solely the utility function that combines simulation costs with solution quality; we provide success rates as an indicator to show the high utility didn't arise from reward-hacking (i.e. proposing highly efficient but inaccurate solutions).
>
> > W4-The experiments are somewhat narrow in scope ...
>
> We would like to emphasize that our study represents one of the first attempts at cost-aware simulation-based experimental design. Even within these simplified settings, our experiments demonstrate that naïve strategies for using LLMs fail (Figure-3; Table-2, Columns 5–6 of our paper). In contrast, the substantial gains achieved by our method not only highlight the promise of CAED-Agent but also illustrate the broader potential of combining lightweight surrogate models, i.e. “world models”, (Tang et al., 2024; Gu et al., 2025) with LLMs in scientific simulation workflows.
>
> Recent work on LLM-automated experiment design (Chan et al., 2025; Chen et al., 2025) implicitly assumes that all forward simulations proposed by LLMs/agents are both reliable and efficient. Our results—even on comparatively simple scenarios—show that this assumption does not hold, especially when the system encounters novel or previously unseen simulation configurations. Thus, we believe our practical contribution lies in revealing the often-overlooked interplay between simulation cost and simulation accuracy in LLM-driven design, and in providing a principled approach for addressing this challenge.
>
> That said, we fully acknowledge the value of more realistic tests. In the revised version, we will include experiments on additional high-complexity systems, such as the Hasegawa–Mima Nonlinear Equation solved with a pseudo-spectral method. This system exhibits strong nonlinear mode coupling and incurs substantial computational cost. Results are currently being produced.
>
> > Q2-How beneficial is it to use a neural network to model the prior knowledge? Would incorporating simple statistics derived from the existing samples (e.g., the possible ranges of the design variables) achieve a similar effect?
>
> We thank the reviewer for a valuable suggestion on ablations. We modified the pdf, adding a paragraph in Section 4.5 under "Using a separate surrogate model for evaluation is more effective than providing samples directly to the LLM." for detailed responses. We showed that few-shot prompting or prompting with statistics to the LLM fail to outperform our method.

---

> > ### Author Response · Authors · 2025-11-22
> > **The added paragraph in Section 4.5 of our updated PDF**
> >
> > **Using a separate surrogate model for evaluation is more effective than providing samples directly to the LLM.**
> >
> > To illustrate necessity of introducing a separately trained neural surrogate to evaluate candidate designs, instead of simply exposing the LLM to the training examples (or summary statistics) through prompting or fine-tuning,  we designed an additional set of experiments. Specifically, we compare the original CAED-Agent against:
> >
> > - Few-shot prompting with 5/10/20 in-context illustrations (Fewshot-5/10/20). As in our original ablation, the illustrations are randomly sampled from samples with the same conditions in the training dataset, arranged in ascending order, and appended to the prompt in Appendix E.
> >
> > - Direct Prompting with statistics derived from all training samples for the neural surrogate. (DP+stats) We provide the following statistics: variable range, best/worst samples, Pearson correlation, fitted model (using quadratic regression) descriptions.
> >
> > - CAED-Agent appended with 5/10/20 in-context illustrations (CAED-Fewshot-5/10/20). The samples are chosen in the same manner as (1), and appended to the prompt. The querying of neural surrogate and iterative update of the illustrations are the same as our original method.
> >
> > | Illustrations     | CAED-Agent | Fewshot | DP+Stats | **CAED-Fewshot** |
> > |-------------------|------------|---------|----------|------------------|
> > | 5 Illustrations   | 0.571      | 0.476   | 0.289    | **0.622**        |
> > | 10 Illustrations  |            | 0.42    |          | **0.635**        |
> > | 20 Illustrations  |            | 0.365   |          | **1.058**        |
> >
> > **Table-1: Experiments done on euler_1d n_space, single-turn.**
> >
> > Results show that (1) Providing only summary statistics consistently underperforms relative to few-shot prompting. Simulation-design tuning is a fine-grained task that requires relational information beyond what statistical descriptors can convey. (2) Few-shot prompting variants also fail to surpass our method, and the variant using an extended context (20 illustrations) performs even worse. Our failure-mode analysis suggests that LLMs tend to copy solutions from illustrations, leading to suboptimal proposals. (3) Augmenting CAED-Agent’s prompts with few-shot illustrations improves the agent’s exploratory behavior, yielding the strongest performance among all tested configurations.

---

### Official Review · Reviewer_rHmV · 2025-10-31

**Soundness:** 2
**Presentation:** 3
**Contribution:** 2
**Rating:** 2
**Confidence:** 4

**Summary:**

The paper introduces CAED-Agent, an agentic framework for simulation-based experimental
design that leverages large language models (LLMs) to optimize simulator configurations
under cost constraints. The central idea is to learn a surrogate model that predicts the
utility (simulation accuracy) and cost (runtime or compute) of running a simulator with
given hyperparameters such as grid size, time step, or solver tolerance. An LLM-based
agent then iteratively proposes new configurations, queries the surrogate for predicted
cost and utility, and updates its prompt to improve the trade-off between fidelity and
efficiency. The authors define single-turn and multi-turn variants of the optimization
process and evaluate CAED-Agent on three physics-based simulators: Heat1D, Euler1D, and
a 2D Navier–Stokes system. They compare against direct LLM prompting, Bayesian
optimization (BO), and the OPRO framework, showing faster convergence and higher reward
in several cases.

**Strengths:**

- The paper explores an emerging and relevant topic: integrating LLMs with surrogate
  modeling for agentic optimization in scientific computing. The combination of cost
  modeling, utility shaping, and multi-turn prompting is original and technically
  interesting.
- The distinction between single-shot design (analogous to standard hyperparameter
  tuning) and multi-round design (analogous to iterative experimental design) is
  conceptually helpful.
- Although limited in scale, the PDE-based testbeds (Heat1D, Euler1D, NS-2D) provide
  reproducible and interpretable environments, which is a strength compared to purely
  synthetic tasks often used in LLM-agent papers.
- Evaluation: The inclusion of Bayesian optimization and OPRO as comparison methods is
  appropriate and helps contextualize the agentic behavior.
- Readable and modular implementation idea: The general pipeline (LLM agent + surrogate
  model + simulator) is easy to grasp and could, in principle, be extended to other
  domains (e.g., actual parameter sweeps, i.e., inference).

**Weaknesses:**

### Problem framing and conceptual clarity

- The problem formulation is ambiguous. The paper presents itself as simulation-based
  experimental design, yet the optimization targets only simulator settings (grid size,
  step size, etc.), not experimental variables or physical parameters. This mismatch
  makes the title and abstract somewhat misleading.
- The introduction jumps directly into the LLM setup without clearly defining what is
  being optimized or why. A reader unfamiliar with the specific simulators will struggle
  to understand the underlying task.
- The notion of “inference-time scaling” is highlighted but never defined. This should
  be introduced more explicitly.
- The downstream purpose of selecting these simulator settings remains unclear. In
  realistic scenarios, one would care about physical inference or control — not just
  choosing a grid size. The authors should motivate how this contributes to actual
  scientific or decision-making goals.

### Relation to prior work and novelty claims

- Overstated novelty in cost modeling. The claim that this is the first approach to
  estimate simulation cost is inaccurate:
- Bharti et al. (2024) explicitly model simulator cost within cost-aware SBI, optimizing
  sampling to minimize total compute.
- Gorecki et al. (2023) address Bayesian decision making via amortized networks, which
  inherently learn expected losses that include cost.
- The paper should clearly position itself relative to these works, correcting the novelty claim.
  - Although Bharti et al. are cited, the connection to SBI remains opaque. Is the
    “utility” function intended to represent inference performance, or is inference
    absent entirely?
  - The decision-making perspective of Gorecki et al. is highly relevant and missing.
    Both approaches learn to choose actions under uncertainty given simulator costs;
    this work could be framed as a heuristic, LLM-driven version of amortized
    decision-making.

### Methodological and conceptual gaps

- The use of an “LLM agent” is poorly formalized. There is no clear notion of an
  optimization objective or theoretical grounding (e.g., expected-utility maximization,
  policy improvement). The surrogate–LLM loop seems heuristic, and no stability or
  convergence analysis is provided.
- The argument that existing experimental design benchmarks cannot be used “because they
  lack cost labels” is weak — costs such as runtime or FLOPs are measurable for any
  simulator. This choice limits comparability to established approaches.
- The tasks are self-contained and synthetic. They demonstrate an internal cost–utility
  trade-off but have no demonstrated downstream relevance or integration with real
  inference workflows.

### Experimental complexity and scalability

- All three PDE benchmarks (1D heat, 1D Euler, 2D Navier–Stokes) are low-dimensional and
  deterministic, with smooth cost–fidelity relations. They are suitable as sanity checks
  but not as evidence of scalability or robustness.
- The paper does not explore increasing complexity (e.g., 3D, chaotic, or stochastic
  regimes). Without such tests, it remains unclear how the method behaves when the
  trade-off surface becomes non-monotonic or discontinuous.
- In more challenging setups, LLM-based optimization is likely to hallucinate or become
  unstable, as it lacks calibrated uncertainty or safety mechanisms. The authors should
  probe these edge cases.

### Missing discussion of uncertainty and trustworthiness

- A major advantage of Bayesian optimization (BO) approaches is that their
  Gaussian-process surrogates provide uncertainty estimates and theoretical guarantees
  for exploration and convergence. The LLM-based approach offers no such calibration, at
  least no principled approaches. Thus, it remains unclear how users can trust or
  interpret its suggestions.
- The paper should explicitly compare BO and LLM agents on a more challenging task,
  evaluating not only final reward but also uncertainty quantification, robustness, and
  failure detection.

### Presentation and writing quality

- Several stylistic and formatting issues (random bolding, missing spaces, inconsistent
  punctuation) reduce readability and suggest unedited LLM-generated text.
- Figures are presented out of order: Figure 1 appears early but is referenced later;
  Figure 2’s caption misstates that the single-turn agent “calls the simulator” rather
  than calling the cost-utility surrogate (unless I am missing something general here?)
- I suggest having Figure 2 as the conceptual Figure 1 early in the paper.

**Questions:**

1. What is the downstream task or use case of optimizing simulator settings? How would
   this framework contribute to a real scientific or decision-making pipeline?
2. How does the method compare in practice to Bayesian optimization on more complex
   problems, particularly in terms of uncertainty estimates and failure detection?
3. Can the surrogate be integrated into a Bayesian decision-theoretic formulation
   similar to Gorecki et al. (2023), allowing direct Bayesian expected-loss minimization
   rather than heuristic LLM guidance?
4. How does the approach scale computationally when the simulator cost becomes dominant
   relative to LLM calls?
5. Have the authors evaluated or observed cases where the LLM proposes invalid or nonsensical
   simulator configurations? How are such failures detected or handled?

---

> ### Author Response · Authors · 2025-11-22
> **Responses to Reviewer rHmV (Part 1)**
>
> We thank the reviewer for the detailed and comprehensive comments.
>
> > W1 - Problem framing and conceptual clarity
>
> > Q1 - What is the downstream task or use case ... a real scientific or decision-making pipeline?
>
> This appears to stem from an insufficient distinction between our problem setting and its downstream application, such as inverse design. We clarify this distinction using the following illustrative example:
>
> Imagine a scenario where an engineer is optimizing the configuration of a long-span bridge. Evaluating a configuration typically requires running a fluid–structure simulation based on multi-dimensional Euler equations. These simulations (Anderson et al., 1995) have numerous design parameters including spatial/temporal resolutions, etc., highly sensitive to boundary conditions and numerical precision. Currently, such parameters are set based largely on the engineer’s intuitions and iterative guesswork.
>
> We refer to the process of inferring the best design configuration as **inverse design**, and the process of evaluating each configuration via simulation as **simulation-based experimentation**. While inverse design is indeed important, its success critically depends on **accurate and cost-efficient simulations** during the design loop, which constitute the focus of our paper.
>
> 1) Imagine if the inverse optimization was conducted based on wrong simulation configurations resulting in bad simulation results, then the validity of the found target will be questionable;
>
> 2) Meanwhile, if every step of simulation is conducted at the finest resolution, the cost will sky rocket, as the inverse design can take up to hundred of simulation runs.
>
> Therefore, How to balance the accuracy-cost trade-off is indeed the key to efficient inverse designs.
>
> The lack of this distinction is also a paradigm of recent work on LLM-automated experiment design (Chan et al., 2025; Chen et al., 2025), where they implicitly assume that forward simulations proposed by LLMs are reliable, thereby focusing primarily on inverse design. This assumption is overly optimistic, indicated by, e.g., the low success rate of LLM conducting a realistic CFD software in Somasekharan er al., (2025). Our experiments also demonstrate this point: the comparisons in Figure 2 show that direct LLM prompting can be both inaccurate and inefficient. These lack of distinction of the problem is our main motivation, and our solution, CAED-Agent, significantly increases success rates and efficiency as a result.
>
> **“Inference-time scaling”** refers to a series of methodologies in LLM-related works that scales up explorations (e.g. with an ensemble) then selecting the best outcome. These methods trade computational resources for an increased performance without training. This term is summarized in survey-style works (Snell et al., 2024; Raschka, 2025).
>
> We agree to revise the abstract, Section 1 and Section 2 of the PDF to include the above clarifications.
>
> [1]Alen Alexanderian, . "Optimal Experimental Design for Infinite-dimensional Bayesian Inverse Problems Governed by PDEs: A Review." (2021).
>
> [2]Jun Shern Chan, , Neil Chowdhury, Oliver Jaffe, James Aung, Dane Sherburn, Evan Mays, Giulio Starace, Kevin Liu, Leon Maksin, Tejal Patwardhan, Lilian Weng, Aleksander Mądry. "MLE-bench: Evaluating Machine Learning Agents on Machine Learning Engineering." (2025).
>
> [3]Ziru Chen, , Shĳie Chen, Yuting Ning, Qianheng Zhang, Boshi Wang, Botao Yu, Yifei Li, Zeyi Liao, Chen Wei, Zitong Lu, Vishal Dey, Mingyi Xue, Frazier N. Baker, Benjamin Burns, Daniel Adu-Ampratwum, Xuhui Huang, Xia Ning, Song Gao, Yu Su, Huan Sun. "ScienceAgentBench: Toward Rigorous Assessment of Language Agents for Data-Driven Scientific Discovery." (2025).
>
> [4]Nithin Somasekharan, , Ling Yue, Yadi Cao, Weichao Li, Patrick Emami, Pochinapeddi Sai Bhargav, Anurag Acharya, Xingyu Xie, Shaowu Pan. "CFDLLMBench: A Benchmark Suite for Evaluating Large Language Models in Computational Fluid Dynamics." (2025).
>
> [5]Charlie Snell, Jaehoon Lee, Kelvin Xu, & Aviral Kumar. (2024). Scaling LLM Test-Time Compute Optimally can be More Effective than Scaling Model Parameters.
>
> [6]Yuichi Inoue, Kou Misaki, Yuki Imajuku, So Kuroki, Taishi Nakamura, & Takuya Akiba. (2025). Wider or Deeper? Scaling LLM Inference-Time Compute with Adaptive Branching Tree Search.
>
> [7] Yangzhen Wu, Zhiqing Sun, Shanda Li, Sean Welleck, & Yiming Yang (2025). Inference Scaling Laws: An Empirical Analysis of Compute-Optimal Inference for LLM Problem-Solving. In The Thirteenth International Conference on Learning Representations.
>
> [8]Raschka, Sebastian (2025). The State of LLM Reasoning Model Inference: Inference-Time Compute Scaling Methods to Improve Reasoning Models.” Ahead of AI Magazine, https://magazine.sebastianraschka.com/p/state-of-llm-reasoning-and-inference-scaling.
>
> [9]Zhi Zheng, , Zhuoliang Xie, Zhenkun Wang, Bryan Hooi. "Monte Carlo Tree Search for Comprehensive Exploration in LLM-Based Automatic Heuristic Design." (2025).

---

> ### Author Response · Authors · 2025-11-22
> **Responses to Reviewer rHmV (Part 2)**
>
> > W2 - Relation to prior work and novelty claims
>
> We thank the reviewer for providing additional reference. We make the following clarifications on our position relative to prior works:
> - We did not claim our method is the first in a generic sense. We have specifically narrated our contribution within LLM-driven experimentation in Related Works. The reviewer’s statement of “over-stated contribution” is not grounded on our writing.
> - We respectfully highlight several key differences from previous SBI works:
>   - **Different definitions of “cost.”** in (Gorecki et al. , 2023): their cost is defined as that of an action (like control action’s effort). In contrast, our cost refers to the complexity of proposed forward simulations.
>   - **Different optimization goals in Bharti et al. (2024).** In this work, SBI’s purpose is to estimate the unknown parameter of simulations (or any forward process) given observations, ie, it’s an inverse design problem. Ours is LLM-driven auto tuning the correct parameters of simulations (or any forward process) to have the simulation done accurately and efficiently. We have distinct this difference in our prior response to W1 and Q1.
>
> Though their problem setting is different and their method is not transferable to our settings directly, the core philosophy of these works, i.e. re-weighting the sampling distribution according to cost, could be combined with BO. We will add this as an ablation of BO during rebuttal. We will also revise our writing and add one more paragraph in Related Works to distinct our differences in the revised manuscript.
>
> > W3 - Methodological and conceptual gaps
>
> For optimization objectives and our potential benefits for downstream applications (such as inverse optimizations), please refer to Section 3.1 of our paper and our response to **W1 and Q1** for a detailed explanation.
>
> Regarding comparability, our statement that “few benchmarks or methods explicitly consider evaluation cost as an optimization target” did not imply their cost cannot be measured. We do not exclude the possibility of evaluating our method on adapted versions of existing benchmarks; however, adapting those benchmarks necessitates knowledge of a problem-specific “cost”, as well as access to the simulator (consequently source code) to add the outputs of “cost measurement”, making such adaptations challenging. This is especially the case when many benchmarks did not open-source data generation, or used commercial software that cannot be modified. Measuring wall time is more general, but fluctuates due to machines, potentially making comparisons inconsistent.
>
> >  W4 - Experimental complexity and scalability
>
> >  W5 - Missing discussion of uncertainty and trustworthiness
>
> > Q6 -How does the method compare in practice to Bayesian optimization ...  uncertainty estimates and failure detection?
>
> For concerns regarding complexity, problem dimensions etc., please refer to **General Responses on Experimental Scenarios** for a detailed explanation. We argue that our results clearly show that naïve LLM-based design strategies fail even in these simple settings, whereas our method achieves substantial gains; we will further strengthen this point with additional experiments on high-complexity systems such as the Hasegawa–Mima equation.
>
> We acknowledge the inevitable hallucination of LLMs, but we argue that (1) our inference-scaling with NN as feedback achieves broad exploration without triggering expensive simulation runs, then the Top-K filtering removes hallucinating proposals. This turns out empirically to have improved performance over all other baselines. (2) Other statistical baselines can not practically avoid failing samples either; the convergence of BO is under strict assumptions of the accuracy of GP and number iterations (Srinivas et al., 2012; Chowdhury & Gopalan, 2017; Bull, 2011), which may not hold true when cost distribution is complex, and the long iteration is too costly.
>
> We agree that this is a valuable future direction, but since it’s challenging to all methods in practice, it’s not a valid reason to reject the contribution of our work.
>
> > W6 - Presentation and writing quality
>
> We thank the reviewer for corrections. An updated version with a compact caption for Figure-2 will be uploaded.
>
> We’ve checked that all bolding and punctuations are intentional; we strictly adhere to our LLM-usage statement, and did not generate paragraphs directly from LLMs.
>
> Figure-1 is put in the beginning intentionally, as an early illustration of our method’s effectiveness over baselines.
>
> [1]Srinivas, Niranjan, Andreas, Krause, Sham M., Kakade, Matthias W., Seeger. "Information-Theoretic Regret Bounds for Gaussian Process Optimization in the Bandit Setting". IEEE Transactions on Information Theory 58. 5(2012): 3250–3265.
>
> [2]Sayak Ray Chowdhury, , Aditya Gopalan. "On Kernelized Multi-armed Bandits." (2017).
>
> [3]Adam D. Bull, . "Convergence rates of efficient global optimization algorithms." (2011).

---

> > ### Author Response · Authors · 2025-11-22
> > **Our General Response on Experimental Scenarios**
> >
> > **Experimental Scenario.**  we would like to emphasize that our study represents one of the first attempts at cost-aware simulation-based experimental design. Even within these simplified settings, our experiments demonstrate that naïve strategies for using LLMs fail (Figure-3; Table-2, Columns 5–6 of our paper). In contrast, the substantial gains achieved by our method not only highlight the promise of CAED-Agent but also illustrate the broader potential of combining lightweight surrogate models, i.e. “world models”, (Tang et al., 2024; Gu et al., 2025) with LLMs in scientific simulation workflows.
> >
> > Recent work on LLM-automated experiment design (Chan et al., 2025; Chen et al., 2025) implicitly assumes that all forward simulations proposed by LLMs/agents are both reliable and efficient. Our results—even on comparatively simple scenarios—show that this assumption does not hold, especially when the system encounters novel or previously unseen simulation configurations. Thus, we believe our practical contribution lies in revealing the often-overlooked interplay between simulation cost and simulation accuracy in LLM-driven design, and in providing a principled approach for addressing this challenge.
> >
> > That said, we fully acknowledge the value of more realistic tests. In the revised version, we will include experiments on additional high-complexity systems, such as the Hasegawa–Mima Nonlinear Equation solved with a pseudo-spectral method. This system exhibits strong nonlinear mode coupling and incurs substantial computational cost. Results are currently being produced.

---

> > > ### Comment · Reviewer_rHmV · 2025-11-26
> > > **Response to rebuttal**
> > >
> > > I thank the authors for their detailed responses to my concerns and questions.
> > >
> > > It was helpful to receive a clarification of the contribution mainly focusing on the optimisation of the simulator configuration and how this is still relevant in practice.
> > > I also appreciate the clarifications being added to the abstract and intro parts.
> > >
> > > Having read the others reviews and the rebuttals, especially the new comparisons to BO, I am wondering about general practical relevance of the method: given that BO actually outperforms CAED in low-budget regimes, how should a practitioner decide when to use CAED?
> > > What are the tradeoffs to be made in terms of computational overhead (also the required hardware for running CAED vs BO) and software tool’s availability (there are plenty of high quality BO frameworks, how is CAED available in terms of OSS?)

---

> > > > ### Author Response · Authors · 2025-11-28
> > > > **Response on Choosing between BO and CAED-Agent**
> > > >
> > > > We thank the reviewer for the timely response and further questions that allow us to continue refining our work, and we appreciate the acknowledgement of our clarifications in the earlier rebuttal.
> > > > > given that BO actually outperforms CAED in low-budget regimes, how should a practitioner decide when to use CAED?
> > > >
> > > > Based on our analysis of BO’s limitations for simulation-based forward experimental design (see our response to **Reviewer 4’s Q1**), we recommend choosing CAED-Agent even under small budgets when the following condition holds:
> > > >  - **The existing samples encode important environmental conditions in natural language** (e.g., “low/medium/high precision,” “sod/lax/mach_3 initial conditions” in our euler_1d setting).
> > > > Given that conventional BO cannot encode these discrete and unordered quantities, they struggles to use utilize this dimension of information:
> > > >      - Treating each condition as a separate BO problem fragments the dataset and drastically reduces the effective sample size for each subtask.
> > > >      - Training a single GP on all data discards the semantic differences among conditions, leading to degraded performance as sample size increases (see our response to Reviewer 1’s W2).
> > > >
> > > > In contrast, CAED-Agent can incorporate these conditions naturally—via one-hot encoding in the surrogate and natural-language prompts for the LLM—providing substantially better initialization and guidance even with limited data.
> > > >
> > > > Otherwise, we suggest a practical alternative: **replace the neural surrogate within CAED-Agent with a GP regressor**. Our preliminary results show that, by substituting the surrogate’s predictions with the GP’s posterior mean and variance, CAED-Agent can achieve performance comparable to BO under low-budget scenarios. In practice, users can therefore alternate between surrogate models (neural networks, GP regressors, or others) based on validation performance within the available training budget. Notably, CAED-Agent can seamlessly adopt different surrogate models because the LLM optimizes using open-form numerical feedback, making the agentic framework inherently surrogate-agnostic.
> > > >
> > > > Complete results are being produced on GP regressor + CAED-Agent, and we would appreciate the reviewer’s confirmation on whether this proposed direction addresses the concern.
> > > >
> > > > >What are the tradeoffs to be made in terms of computational overhead (also the required hardware for running CAED vs BO) and software tool’s availability (there are plenty of high quality BO frameworks, how is CAED available in terms of OSS?)
> > > >
> > > > **Computational overhead.** We first provide an estimation of the training and inference costs for the GP regressor (used in BO) and the neural surrogate (used in CAED-Agent):
> > > >  - Training cost: A standard GP regressor requires computing and inverting the $n \times n$ kernel matrix:
> > > > $$\mathbf{K} \in \mathbb{R}^{n \times n}, \qquad
> > > > K_{ij} = k(\mathbf{x}_i, \mathbf{x}_j), \quad i,j = 1,\dots,n.$$
> > > >     This implies: Kernel matrix construction: $O(n^2 d)$ and Matrix inversion / Cholesky decomposition: $O(n^3)$, scaling the   total training cost to $\boxed{O(n^3)}$. Whereas for NNs with P total parameters, one epoch over the entire dataset costs     O(nP), and the total training cost if trained for E epochs is: $\boxed{O(E n P)}$.
> > > > - Inference cost: For GP regressor, predicting at a new test point $x_\*$ requires: computing kernel vector $k(x_\*, X): O(n d)$ and multiplying by the stored Cholesky inverse: $O(n)$, bringing total cost to $\boxed{O(n)}$. For neural network, a simple forward pass is only $\boxed{O(P)}.$
> > > >
> > > > As the number of samples increases, the computational burden of GP fitting grows rapidly and soon becomes prohibitive, while NN training remains scalable. Additional overhead (BO’s acquisition computation or CAED-Agent’s LLM API calls) is negligible compared with the above costs.
> > > >
> > > > **Software Availability**:  We stress that CAED-Agents consists of the neural network training module and the LLM agent module, each supported by mature, widely used open-source ecosystems as well:
> > > >  - Neural network training frameworks: PyTorch, JAX / Flax, Lightning
> > > >  - LLM agent and tool-execution frameworks: LangChain (we used for our implementation), AutoGen (Microsoft), OpenAI function calling / tool API.
> > > >
> > > > Thus, CAED-Agent can be implemented and extended entirely using existing, well-supported OSS tooling.

---

> > > > > ### Comment · Reviewer_rHmV · 2025-11-28
> > > > >
> > > > > I thank the reviewers for the timely response to my questions.
> > > > >
> > > > > Yes, the multimodality of the LLM approach compared to BO is a point an agree with.
> > > > >
> > > > > Overall, I am now more convinced of this contribution and will adapt my score accordingly.
> > > > >
> > > > > For the updated version of the manuscript I think it is important to make the initial framing of the problem (focus on simulator configuration vs design / parameter inference) very clear in order to avoid any misinterpretation by the reader.
> > > > >
> > > > > Additionally, I find it very important to add these practical considerations you explained in your last two comments to the main paper as well (potentially in a briefer format) to give practitioners some guidance, including links to the code repository for reproducing the methods and results of this paper.
> > > > >
> > > > > Thank you for the discussion!

---

### Official Review · Reviewer_zDSZ · 2025-10-31

**Soundness:** 2
**Presentation:** 3
**Contribution:** 2
**Rating:** 4
**Confidence:** 3

**Summary:**

The authors present a method to optimally select parameters for computationally complex computer simulations, comprised of a learned surrogate that models the cost and performance of different parameters and an agentic LLM that queries the surrogate to optimize the parameters.

**Strengths:**

The idea of training a neural network surrogate for an agentic LLM to query for optimizing simulations is interesting and novel. It is useful to be able to integrate prior knowledge into the optimization process.

**Weaknesses:**

The authors do not present a convincing argument that CAED-agent is consistently better than the state of the art. It is unclear what hyperparameters were used to generate the results for the baseline experiments. It is also unclear if a fair comparison was made, or in what dimension the baseline metrics were equivalent to the CAED-agent metrics. Did the methods have equivalent runtimes, equivalent queries of the simulator, or equivalent computational requirements?

How does performance for each method change with different constraints? For example, BO is usually sample efficient and would likely still produce reasonable results with a few dozen queries of the simulator, but I would imagine the surrogate neural network would not be able to train on just a few dozen queries. Can the authors run an ablation on these parameters?

**Questions:**

If you have the neural proxy for the true simulator, does BO work on the neural proxy? On the other hand, is giving the LLM the training data directly and asking for an optimized result possibly more effective than using the data to train a neural proxy then having a separate LLM to query the proxy?

---

> ### Author Response · Authors · 2025-11-22
> **Responses to Reviewer zDSZ**
>
> We sincerely thank the reviewer for recognizing the contributions of our work and for the careful evaluation and constructive comments and questions.
>
> > W1- It is unclear what hyperparameters were used ... equivalent runtimes, equivalent queries of the simulator, or equivalent computational requirements?
>
> We thank the reviewer for pointing this out. We have provided implementation details, hyper-parameters and computational budgets in Appendix G of the updated pdf.
>
> > W2-How does performance for each method change with different constraints? .... Can the authors run an ablation on these parameters?
>
> We thank the reviewer for this helpful suggestion regarding additional ablations. For clarity, we first restate our implementation of BO: we fit a Gaussian Process regressor on the same training dataset used to train the neural surrogates, and allow either 1 or 10 queries to the true simulator for the acquisition function, depending on whether the setting is single-turn or multi-turn.
>
> We show the required ablation results below. Specifically, we compare
> - CAED-Agent with neural surrogate trained on 50, 100, 500 or full (~4k) samples.
> - BO with GP regressor fitted on 50, 100, 500 or full (~4k) samples.
>
> | Method              | 50 samples | 100 samples | 500 samples | full samples (~4k) |
> |---------------------|------------|-------------|-------------|---------------------|
> | **CAED-Agent (Ours)** | 0.142      | 0.239       | 0.471       | **0.571**           |
> | BO                  | 0.459      | 0.418       | 0.392       | 0.391               |
>
> **Table-1: Ablation on training dataset size for the GP regressor (BO) and neural surrogate (CAED-Agent). Experiments conducted on euler_1d n_space, single-turn.**
>
> Our results show that 1)  BO does outperform our method when only very small datasets (fewer than 100 samples) are available, however 2) BO’s performance downgrade with more samples and 3) CAED-Agent outperforms BO once training data exceeds 500 samples.These findings indicate when sufficient computational resources are available for offline data accumulation, or if the lab already has abundant logs of recordings, our current approach has higher upper bound. Still, we also realize under low data regimes, GP+BO can be a better alternative and even combined with our current framework (eg, agent chooses which surrogate+optimization scheme to use), hence will add it as a future direction.
>
> > Q1-If you have the neural proxy for the true simulator, does BO work on the neural proxy? On the other hand, is giving the LLM the training data directly and asking for an optimized result possibly more effective than using the data to train a neural proxy then having a separate LLM to query the proxy?
>
> We thank the reviewer for valuable suggestions on ablation. Regarding "giving the LLM the training data directly", we modified the pdf, adding a paragraph in Section 4.5 under "Using a separate surrogate model for evaluation is more effective than providing samples directly to the LLM." for detailed responses. We showed that few-shot variants that provide samples directly to the LLM fail to outperform our method—and even degrade when using extended context—whereas augmenting CAED-Agent prompts with few-shot illustrations enhances exploration and yields the best overall performance.
>
> For your suggestion on optimizing the neural proxy with BO, we provide the following results:
> we compare
>
> 1. **CAED-Agent:** optimizing the neural proxy with LLM
> 2. **BO on GP:** optimizing actual simulator output with BO
> 3. **BO on neural proxy (BO on NN):** BO is used to optimize the neural proxy’s reward output with 10 queries (to enable this, we use random drop-out and ensemble to estimate NN’s uncertainty), then evaluate the final result with the actual simulator.
>
> ---
>
> ### Results
>
> | Metric | CAED-Agent (Ours) | BO on GP | BO on NN |
> |--------|------------|----------|-----------|
> | Reward | 0.571      | 0.391    | 0.290     |
>
> ---
>
> Our results show that BO on NN underperforms both CAED-Agent and standard BO on GP. This may be due to BO relies on accurate uncertainty estimation which GP inherently excels. The uncertainty estimation of NN however is harder, containing heuristics and parameter tuning, making BO on NN not efficient.

---

> > ### Author Response · Authors · 2025-11-22
> > **The added paragraph in Section 4.5 of our updated PDF**
> >
> > **Using a separate surrogate model for evaluation is more effective than providing samples directly to the LLM.**
> >
> > To illustrate necessity of introducing a separately trained neural surrogate to evaluate candidate designs, instead of simply exposing the LLM to the training examples (or summary statistics) through prompting or fine-tuning,  we designed an additional set of experiments. Specifically, we compare the original CAED-Agent against:
> >
> > - Few-shot prompting with 5/10/20 in-context illustrations (Fewshot-5/10/20). As in our original ablation, the illustrations are randomly sampled from samples with the same conditions in the training dataset, arranged in ascending order, and appended to the prompt in Appendix E.
> >
> > - Direct Prompting with statistics derived from all training samples for the neural surrogate. (DP+stats) We provide the following statistics: variable range, best/worst samples, Pearson correlation, fitted model (using quadratic regression) descriptions.
> >
> > - CAED-Agent appended with 5/10/20 in-context illustrations (CAED-Fewshot-5/10/20). The samples are chosen in the same manner as (1), and appended to the prompt. The querying of neural surrogate and iterative update of the illustrations are the same as our original method.
> >
> > | Illustrations     | CAED-Agent | Fewshot | DP+Stats | **CAED-Fewshot** |
> > |-------------------|------------|---------|----------|------------------|
> > | 5 Illustrations   | 0.571      | 0.476   | 0.289    | **0.622**        |
> > | 10 Illustrations  |            | 0.42    |          | **0.635**        |
> > | 20 Illustrations  |            | 0.365   |          | **1.058**        |
> >
> > **Table-1: Experiments done on euler_1d n_space, single-turn.**
> >
> > Results show that (1) Providing only summary statistics consistently underperforms relative to few-shot prompting. Simulation-design tuning is a fine-grained task that requires relational information beyond what statistical descriptors can convey. (2) Few-shot prompting variants also fail to surpass our method, and the variant using an extended context (20 illustrations) performs even worse. Our failure-mode analysis suggests that LLMs tend to copy solutions from illustrations, leading to suboptimal proposals. (3) Augmenting CAED-Agent’s prompts with few-shot illustrations improves the agent’s exploratory behavior, yielding the strongest performance among all tested configurations.

---

### Author Response · Authors · 2025-11-22
**General Responses**

We thank the reviewers for their careful evaluation of our work and their constructive comments. We would like to first address several common concerns.

**Experimental scenario.** Many reviewers raised questions regarding the dimensionality, complexity, and diversity of our simulators. For example:

>R2W - All three PDE benchmarks (1D heat, 1D Euler, 2D Navier–Stokes) are low-dimensional and deterministic, ... scalability or robustness.

>R3W - The experiments are somewhat narrow in scope ... a small number of design variables

>R4W - the evaluation is confined to toy problems; ... high-dimensional or noisy.

While these observations are factually correct, we would like to emphasize that our study represents one of the first attempts at **cost-aware simulation-based experimental design**. Even within these simplified settings, our experiments demonstrate that naïve strategies for using LLMs fail (Figure-3; Table-2, Columns 5–6). In contrast, the substantial gains achieved by our method not only highlight the promise of CAED-Agent but also illustrate the broader potential of combining surrogate models, or “world models”, (Tang et al., 2024; Gu et al., 2025) with LLMs  in scientific simulation workflows.

Recent work on LLM-automated experiment design (Chan et al., 2025; Chen et al., 2025) implicitly assumes that all forward simulations proposed by LLMs/agents are both reliable and efficient. Our results—even on comparatively simple scenarios—show that this assumption does not hold, especially when the system encounters novel or previously unseen simulation configurations. Thus, we believe our practical contribution lies in revealing the often-overlooked interplay between simulation cost and simulation accuracy in LLM-driven design, and in providing a principled approach for addressing this challenge.

That said, we fully acknowledge the value of more realistic tests. In the updated version, we will include experiments on additional high-complexity systems, such as the **Hasegawa–Mima Nonlinear Equation** solved with a pseudo-spectral method. This system exhibits strong nonlinear mode coupling and incurs substantial computational cost. Results are currently being produced.

**Using a separate neural surrogate for evaluation.** Several reviewers questioned the necessity of introducing a separately trained neural surrogate to evaluate candidate designs, instead of simply exposing the LLM to the training examples (or summary statistics) through prompting or fine-tuning, such as:

> R1Q - If you have the neural proxy ... a separate LLM to query the proxy?

> R4W - The design choice of adding a small neural network ... theoretical or empirical backing.

> R3Q - How beneficial is it to use a neural network ... achieve a similar effect?

 We have added **a paragraph in Section 4.5 of the updated pdf** under "Using a separate surrogate model for evaluation is more effective than providing samples directly to the LLM." for detailed responses. We showed that methods directly providing samples to the LLM fail to outperform our method—and even degrade when using extended context—whereas augmenting CAED-Agent prompts with few-shot illustrations enhances exploration and yields the best overall performance.

**Implementation details and experimental settings.** We have provided implementation details, hyper-parameters and computational budgets in **Appendix G of the updated pdf.**


[1]Hao Tang, , Darren Key, Kevin Ellis. "WorldCoder, a Model-Based LLM Agent: Building World Models by Writing Code and Interacting with the Environment." (2024).

[2]Yu Gu, , Kai Zhang, Yuting Ning, Boyuan Zheng, Boyu Gou, Tianci Xue, Cheng Chang, Sanjari Srivastava, Yanan Xie, Peng Qi, Huan Sun, Yu Su. "Is Your LLM Secretly a World Model of the Internet? Model-Based Planning for Web Agents." (2025).

[3]Jun Shern Chan, , Neil Chowdhury, Oliver Jaffe, James Aung, Dane Sherburn, Evan Mays, Giulio Starace, Kevin Liu, Leon Maksin, Tejal Patwardhan, Lilian Weng, Aleksander Mądry. "MLE-bench: Evaluating Machine Learning Agents on Machine Learning Engineering." (2025).

[4]Ziru Chen, , Shĳie Chen, Yuting Ning, Qianheng Zhang, Boshi Wang, Botao Yu, Yifei Li, Zeyi Liao, Chen Wei, Zitong Lu, Vishal Dey, Mingyi Xue, Frazier N. Baker, Benjamin Burns, Daniel Adu-Ampratwum, Xuhui Huang, Xia Ning, Song Gao, Yu Su, Huan Sun. "ScienceAgentBench: Toward Rigorous Assessment of Language Agents for Data-Driven Scientific Discovery." (2025).

---

### Author Response · Authors · 2025-12-04
**Messages to AC**

To Area Chair,

In light of the recent adjustment to reviewing mechanisms, we’d like to provide a summary of the rebuttal up until now.

 - Several requests for more detailed experimental settings to show whether a fair comparison has been made were promptly addressed with an added **Appendix G and H** in the pdf.

 - In response to multiple reviewers’ requests, we added ablation results comparing in-context learning with our approach. Results show that in-context learning (providing few-shot illustrations or summary of statistics to the LLM) fails to surpass our method, which trains a separate neural network to provide feedback.

 - Questions on sampling efficiency relative to Bayesian Optimization were addressed from two angles: (a) our method is more potential than BO when sufficient samples are available, and (b) our method is flexible with respect to surrogate choice, allowing a switch to more sample-efficient surrogates when training samples are limited. As shown in the new results in Section 4.5, integrating our method with a Gaussian Process surrogate enables it to surpass BO in low-sample regimes.

 - We also implemented several presentation-related suggestions by revising the title, abstract, introduction, related works, and Figure 2. In particular, we renamed our problem setting to “Cost-Aware Simulation Configuration Optimization” to avoid misunderstandings such as the one raised by Reviewer 2.

We also note that, after our initial rebuttal messages, **reviewer 2 raised their rating to 4**. After we addressed their follow-up concerns, reviewer 2 explicitly acknowledged that they were **“now more convinced of this contribution and will adapt [their] score accordingly.”**

We have addressed most concerns regarding presentation, experimental setting, ablations, etc. Therefore, we kindly encourage the Area Chair to take our other three dedicated (yet unanswered) rebuttal messages into consideration. Thanks!

---

### Meta-Review · Area_Chair_PLGu · 2026-01-05

**Summary:**

The paper proposes CAED-Agent, an LLM-driven agent for cost-aware simulation configuration optimization, where a learned surrogate predicts utility and cost and the LLM iteratively proposes simulator settings under budget constraints.

Reviewers agreed the problem is relevant and the surrogate+agent loop is interesting, but raised concerns about
- (1) ambiguous framing (experimental design vs simulator hyperparameter tuning),
- (2) fairness/completeness of baselines and experimental details, and
- (3) limited realism/scale of the PDE testbeds and dependence on sizable offline data for training the surrogate. The rebuttal added missing details and several targeted ablations, and at least one initially negative reviewer indicated they are now more convinced and would raise their score.

**Reviewer Concerns:**

Addressed by rebuttal:
- Experimental details / fairness: Added appendices with hyperparameters and budgets, reducing “unclear comparison” concerns.
- Why a separate surrogate vs in-context learning: New ablations show direct prompting / few-shot / stats generally underperform, while the surrogate-guided agent does better; augmenting CAED with few-shot can help further.
- Low-budget vs BO: Added ablations over dataset size showing BO can win in very low-sample regimes, but CAED surpasses once enough data is available; also tried BO-on-NN and showed it underperforms.
- Framing confusion: Authors explicitly rename and clarify the setting as simulator configuration optimization and commit to revising title/abstract/intro accordingly; reviewer feedback suggests this helped.

Still outstanding:
- Practical relevance and scope: The evaluation remains on relatively small, mostly deterministic PDE setups; “more complex system results are being produced” is not evidence yet.
- When to use CAED vs BO in practice: The rebuttal gives a plausible story (heterogeneous/natural-language conditions, surrogate-agnostic framework), but the main paper likely still needs a short, concrete decision guideline and clear reporting of overheads (LLM calls, training cost).
- Positioning vs related work: Some novelty/positioning concerns remain (cost-aware SBI / decision-theoretic viewpoints, and BO variants with priors). The rebuttal addresses this, but the camera-ready must tighten it.

**Reviewer Scores:**

If reviewers had fully participated:
- rHmV: 2 → 4–5 (they explicitly say they will adapt their score upward after rebuttal).
- zDSZ: 4 → 4 (concerns largely addressed, but likely still borderline).
- WNEs: 4 → 4 (ablation and details help; scope/data regime concerns remain).
- toXH: 4 → 4 (same: improved clarity/ablations, but scalability remains open).

---

### Decision · Program_Chairs · 2026-01-26

Reject